# SceneDesigner: Controllable Multi-Object Image Generation with 9-DoF Pose Manipulation

**Zhenyuan Qin**[*]   **Xincheng Shuai**[*]   **Henghui Ding**[†]
**Fudan University**
`https://github.com/FudanCVL/SceneDesigner`

## Abstract

Controllable image generation has attracted increasing attention in recent years, enabling users to manipulate visual content such as identity and style. However, achieving simultaneous control over the 9D poses (location, size, and orientation) of multiple objects remains an open challenge. Despite recent progress, existing methods often suffer from limited controllability and degraded quality, falling short of comprehensive multi-object 9D pose control. To address these limitations, we propose *SceneDesigner*, a method for accurate and flexible multi-object 9-DoF pose manipulation. *SceneDesigner* incorporates a branched network to the pre-trained base model and leverages a new representation, CNOCS map, which encodes 9D pose information from the camera view. This representation exhibits strong geometric interpretation properties, leading to more efficient and stable training. To support training, we construct a new dataset, *ObjectPose9D*, which aggregates images from diverse sources along with 9D pose annotations. To further address data imbalance issues, particularly performance degradation on low-frequency poses, we introduce a two-stage training strategy with reinforcement learning, where the second stage fine-tunes the model using a reward-based objective on rebalanced data. At inference time, we propose *Disentangled Object Sampling*, a technique that mitigates insufficient object generation and concept confusion in complex multi-object scenes. Moreover, by integrating user-specific personalization weights, *SceneDesigner* enables customized pose control for reference subjects. Extensive qualitative and quantitative experiments demonstrate that *SceneDesigner* significantly outperforms existing approaches in both controllability and quality.

## 1   Introduction

Controlling the spatial properties of real-world images has been extensively explored, enabling users to manipulate the structure of interesting subjects or the overall layout of a scene [65, 58, 23, 18, 16, 33, 34, 6, 54]. However, most existing methods are confined to 2D space and often rely on densely annotated control maps as input, such as depth images. In contrast, 3D spatial control remains a largely underexplored challenge. Consider, for example, a designer aiming to arrange multiple pieces of furniture in a room, each with distinct sizes and orientations, or a user wishing to generate an image where a pet dog is turned away from the camera, gazing at the landscape ahead. These scenarios highlight the need for generation models that support 3D-aware multi-object control, a capability that is both essential for practical applications and insufficiently addressed by current approaches.

There have been several preliminary explorations in 3D-aware controllable generation [38, 32, 28, 17, 8, 25, 48]. For example, LOOSECONTROL [6] employs 3D bounding boxes for controlling the

---

[*]Equal Contribution

[†]Henghui Ding (henghui.ding@gmail.com) is the corresponding author with the Institute of Big Data, College of Computer Science and Artificial Intelligence, Fudan University, Shanghai, China.

39th Conference on Neural Information Processing Systems (NeurIPS 2025).

**(a) Single-object 9D Pose Control**

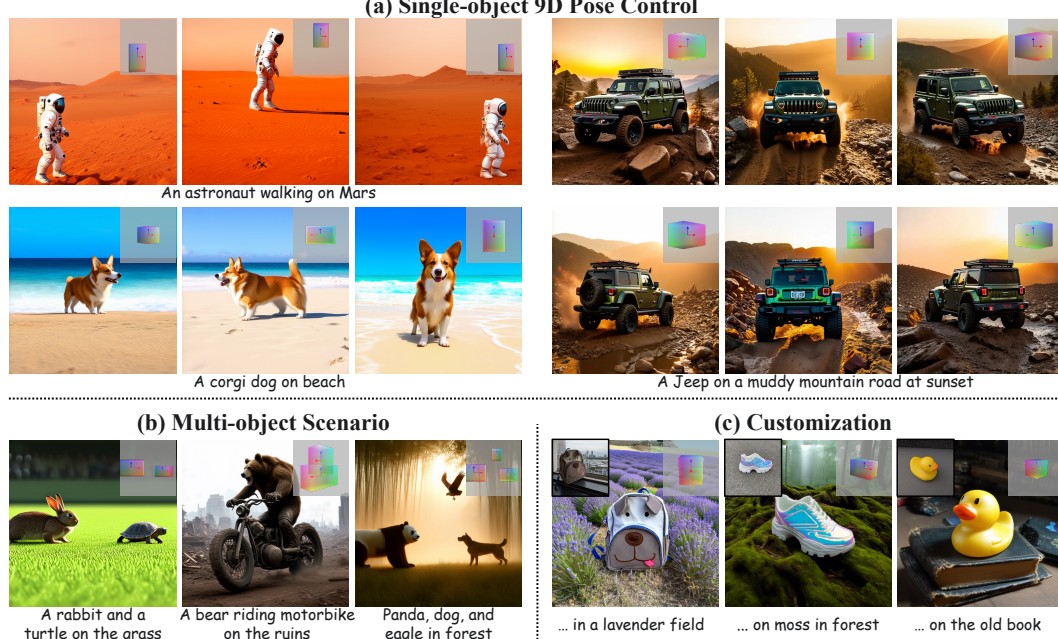

An astronaut walking on Mars

A corgi dog on beach

A Jeep on a muddy mountain road at sunset

**(b) Multi-object Scenario**

A rabbit and a turtle on the grass

A bear riding motorbike on the ruins

Panda, dog, and eagle in forest

**(c) Customization**

… in a lavender field

… on moss in forest

… on the old book

Figure 1: 9D pose control results of the *SceneDesigner*. The figures show the applications in single-object, multi-object, and customization scenarios, exhibiting high quality, flexibility and fidelity.

location and size of the object in 3D space. To enable orientation control, some methods take rotation angles around designated axes as input. Among them, Zero-1-to-3 [28] infers the novel perspective of reference subject, but relies on external inpainting tools [44] to create a complex background. Continuous 3D Words [8] and its following work [38] are constrained by unrealistic visual style and limited control over object quantity and pose diversity. More recently, ORIGEN [32] introduces precise orientation control but is restricted by its dependence on a one-step generative model, which hinders compatibility with widely used multi-step frameworks [44, 39, 10].

To address the aforementioned challenges, we enhance existing text-to-image models [10, 44] by enabling 9D pose control of multiple objects within the same scene, as shown in Fig. 1. To support this goal, we first construct a new dataset, *ObjectPose9D*, which provides 9D pose annotations across diverse real-world scenarios. To build *ObjectPose9D*, we begin with the publicly available OmniNOCS dataset [20], which offers accurate pose annotations but is limited in object and background diversity. To overcome this limitation, we further annotate the large-scale MS-COCO dataset [26] with 9D poses to expand the variety of visual concepts and scene types. Specifically, we employ MoGe [57] and Orient Anything [59] to estimate 3D bounding boxes with orientations. All collected images and pose annotations are then carefully checked and manually refined by human annotators to ensure data quality. Together, these efforts yield *ObjectPose9D*, a diverse and richly annotated dataset for training image generation models with flexible multi-object 9D pose control.

Then, *how can 9D poses be efficiently encoded for controllable image generation?* Some previous studies [8, 38] project rotation angle into textural space and combine it with text embeddings. However, this way struggles to capture fine-grained orientation and precise spatial positioning. Inspired by the success of ControlNet-like architectures [65, 67, 41, 34] in structural

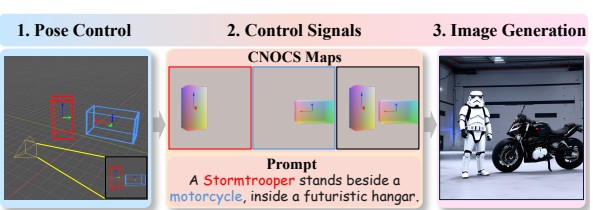

Figure 2: Overview of *SceneDesigner*.

control, LOOSECONTROL [6] adopts 3D bounding boxes for 3D-aware guidance. However, this representation lacks orientation information. For example, a single bounding box may ambiguously correspond to either a front- or back-facing object, limiting controllability and reliability. To address this, we build upon the Normalized Object Coordinate System (NOCS) [52] to better encode geometric properties. Nevertheless, a traditional NOCS map requires a precise 3D shape of each object category, which is user-unfriendly and often difficult to acquire. To overcome this, we introduce the Cuboid NOCS (CNOCS) map to only consider a cuboid shape for general purpose. This simplification retains essential geometric cues while supporting category-agnostic pose encoding. Moreover,

CNOCS map allows for flexible variants and generalizes well across object categories. Besides, to address the imbalanced pose distribution in real-world data, we introduce a two-stage training strategy with reinforcement learning. In the first stage, the model is trained on *ObjectPose9D* to learn basic pose controllability. In the second stage, it is fine-tuned to improve performance on low-frequency object poses by maximizing our proposed reward function under a balanced distribution. During inference, we apply *Disentangled Object Sampling*, a technique designed to mitigate concept fusion and insufficient generation in complex multi-object scenes. Furthermore, by integrating user-specific personalized weights, our method enables customized pose control for user-provided reference subjects. Within our framework, users can freely operate the cuboid-shape meshes in 3D space as demonstrated in Fig. 2, identifying the 3D location, size, and orientation of each object.

Our main contributions are: **1)** We introduce *ObjectPose9D*, a dataset with rich real-world scenes and comprehensive 9D pose annotations, facilitating effective training of pose-controllable generation models. **2)** We propose *SceneDesigner*, a framework that supports multi-object 9D pose control. It leverages our proposed pose representation, CNOCS map, which encodes 3D properties of objects from the camera view using a coarse cuboid abstraction. We further introduce a two-stage training strategy with reinforcement learning to mitigate data imbalance and enhance generalization. At inference time, *Disentangled Object Sampling* is employed to address insufficient or entangled object generation, while user-specific weights enable personalized pose control. **3)** Extensive qualitative and quantitative experiments demonstrate that *SceneDesigner* significantly outperforms previous methods in both single- and multi-object scenarios, achieving high fidelity and controllability.

## 2 Related Works

**Controllable generation:** Controllable image generation [65, 67, 41, 34, 63, 24, 55, 45, 13, 2, 4, 21, 35, 47] enables fine-grained user control over visual content. Among them, ControlNet [65] and its following works [41, 67] introduce a branched network to process the geometry guidance, resulting in the image with high structure fidelity. Other methods [2, 4] like DreamBooth [45] and Textual Inversion [13] learn new concepts given by users, endowing the text-to-image (T2I) models with customization capability. Despite these efforts, controlling the 9D poses of objects still faces challenges. Most methods [23, 58, 61] can only handle 2D bounding boxes, which are insufficient to represent 3D properties. Recently, LOOSECONTROL [6] lifts this condition to 3D space, but struggles to represent precise orientation. A group of methods [38, 8] leverage precise pose annotations from a synthetic dataset, and receive the rotation angles from users. However, they are constrained by limited controllability in intricate pose and multi-object scenarios, and poor generalization in in-the-wild images. The recent work ORIGEN [32] optimizes initial noise to maximize the constructed reward function. However, it cannot localize objects in designated area. Additionally, it relies on one-step generation models, hampering its compatibility with general models. Collectively, there are currently no approaches that can efficiently control the 9D poses of multiple objects in image generation. Beyond the image domain, recent advances in video generation have explored 3D-aware controllable synthesis. 3DTrajMaster [12] focuses on multi-entity trajectory control through MLP-encoded pose representations, while FMC [48] enables 6D pose control for cameras and objects. Meanwhile, CineMaster [54] extends the LOOSECONTROL [6] paradigm to video synthesis, providing intuitive cinematic control.

**3D-aware image editing:** There is a growing body of methods [66, 28, 56, 64, 37, 21] that focus on 3D-aware image editing. Among them, 3DIT [31] was trained on collected synthetic data, manipulating the object by translation or rotation. Similarly, Zero-1-to-3 [28] can generate novel views of the reference subject given by users. NeuralAssets [60] learns object-centric representations that enable disentangled control over object appearance and pose in neural rendering pipelines. Different from these methods, Diffusion Handles [37] leverages the point clouds estimated from the input image, and guides the sampling process through transformed points and depth-conditioned ControlNet [65]. 3DitScene [66] reconstructs the 3D Gaussian Splatting [19] of the scene, and manipulates the 3D Gaussians of the designated object through arbitrary 3D-aware operations.

**Aligning the generation model with human preference:** Based on the success of reinforcement learning in the field of natural language processing [36, 43], some studies [11, 7] make efforts to align the generated image with human preference. Some methods [62, 40] like ReFL [62] maximizes the proposed reward score through backpropagation. Other RL-based methods like DDPO [7] leverage

policy gradient algorithm for finetuning. Different from these methods, Diffusion-DPO [51] performs supervised learning on constructed preference dataset.

## 3 Method

### 3.1 Preliminaries

**Generative models:** The generative models are designed to transform samples from a simple noise distribution into data from the target distribution. Diffusion-based methods [49, 50, 14] learn a noise prediction model, and remove the estimated noise from noisy image during sampling. From another perspective, Flow matching [3, 27, 30] achieves the goal by training a neural network to model the velocity field, moving the noise to data along straight trajectories.

**Normalized Object Coordinate System (NOCS):** To estimate 6D poses and sizes of multiple objects, NOCS [52] constructs the correspondences between pixels and normalized coordinates. It represents all objects in a normalized space while maintaining consistent category-level orientation. Then, the method constructs NOCS map, an RGB image representing each pixel's normalized coordinates, which is efficient for training the prediction model.

### 3.2 Overview

Our goal is to equip T2I models [44, 39, 10] with the controllability in 9D poses of multiple objects. For clarity, we only present our flow-based implementation [10]. It's worth noting that our method can also be applied to diffusion models [44, 39]. Formally, given the text prompt $c_p$ depicting the visual content that contains a set of objects $\{\text{obj}_i\}_{i=1}^{N_o}$, and their 9D poses $\{l_i, s_i, o_i\}_{i=1}^{N_o}$, where $l_i, s_i, o_i$ represent the 3D location, size, and orientation from the camera view, respectively. The generation model aims to create the final image $x$ that minimizes $\Sigma_{i=1}^{N_o} \big( D_{ls}(x, \text{obj}_i, l_i, s_i) + D_o(\mathcal{X}(x, \text{obj}_i), o_i) \big)$, where $D_{ls}$ indicates the inaccuracy of location and size in image space. $\mathcal{X}$ infers the object's orientation and $D_o$ is a discriminant function that calculates the discrepancy of input and estimated orientations. For simplicity, we denote the $\{l_i, s_i, o_i\}_{i=1}^{N_o}$ as $\{\mathcal{P}_i\}_{i=1}^{N_o}$. Therefore, our task is to learn a conditional generation model $p_\theta(x|c_p, \{\mathcal{P}_i\}_{i=1}^{N_o})$ parameterized by $\theta$ that minimizes:

$$E_{t,(x,c_p,\{\mathcal{P}_i\}_{i=1}^{N_o})\sim\mathcal{D},\epsilon\sim\mathcal{N}(\mathbf{0},\mathbf{I})}\left[\left\|v_\theta\left(x_t,t,c_p,\{\mathcal{P}_i\}_{i=1}^{N_o}\right)-(\epsilon-x)\right\|^2\right],\tag{1}$$

where $v_\theta$ models velocity field, the data are sampled from *ObjectPose9D*, denoted as $\mathcal{D}$, and $x_t = (1-t)x + t\epsilon$. In following sections, Sec. 3.3 introduces our CNOCS map that encodes multi-object 9D poses $\{\mathcal{P}_i\}_{i=1}^{N_o}$. Then, Sec. 3.4 details the construction pipeline of our dataset *ObjectPose9D*. Finally, Sec . 3.5 presents our two-stage training strategy for enhancing the alignment with input condition, and *Disentangled Object Sampling* technique used at inference time.

### 3.3 CNOCS Map: Effective Representation of 9-DoF poses

To encode the 9D poses of general objects, *i.e.*, $\{\mathcal{P}_i\}_{i=1}^{N_o}$, a straightforward method is to project the location, size, and orientation to embeddings, respectively, while integrating the features into network through cross- or self-attention mechanism [8, 38, 58, 23, 53, 22]. In contrast, control maps used in ControlNet-like methods [65, 41] offer an alternative approach for encoding 9D pose information. This spatial representation provides stronger structural constraints compared to direct projection methods. Our subsequent experimental results (Sec. 4) demonstrate that this map-based representation is more effective than direct embedding approaches, therefore, we adopt this encoding strategy for our framework.

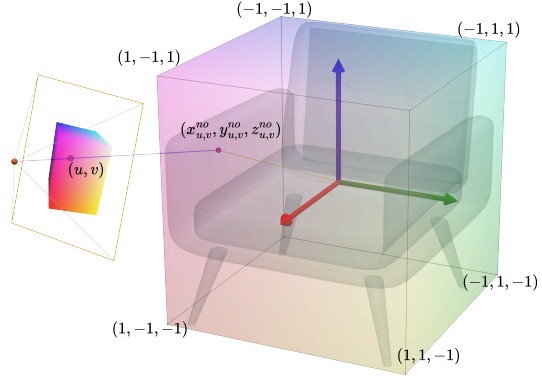

Figure 3: Illustration of CNOCS map.

We leverage the idea from NOCS [52] and propose CNOCS map to preserve the object's 3D properties, exhibiting strong geometry interpretation. A preliminary idea is to obtain the 3D bounding boxes of objects and render them in depth-sorted order depending on their distances from the camera view, which have already been explored in recent works [6]. Although this kind of representation can perceive the location and size of objects, it is viewpoint-dependent and insufficient to present precise orientation. Inspired by NOCS, it's necessary to establish correspondence between pixels indexed by $(u, v)$ and their associated points on the object's surface, while encoding the point based on its coordinate in object space. However, NOCS has to access the precise 3D shape of each category, leading to cumbersome user input and hindering its application. In contrast, our CNOCS map alleviates the issues by using a cuboid shape inherited from 3D bounding boxes, while formalizing the encoding process to achieve better generality. The process of constructing CNOCS map is illustrated in Fig. 3 and formalized in Eq. 2.

$$
\begin{aligned}
x_{u,v}^c, y_{u,v}^c, z_{u,v}^c, \mathrm{oi} &= \mathrm{Intersect}(u, v, \{\mathrm{bbox}_i\}_{i=1}^{N_o}), \\
x_{u,v}^o, y_{u,v}^o, z_{u,v}^o &= \mathrm{proj}_{\mathrm{c2o}}^{\mathrm{oi}}(x_{u,v}^c, y_{u,v}^c, z_{u,v}^c), \\
x_{u,v}^{no}, y_{u,v}^{no}, z_{u,v}^{no} &= \mathrm{normalize}(x_{u,v}^o, y_{u,v}^o, z_{u,v}^o, \mathrm{bbox}_{\mathrm{oi}}), \\
\mathrm{CNOCS}[u, v] &= f(x_{u,v}^{no}, y_{u,v}^{no}, z_{u,v}^{no}),
\end{aligned}
\tag{2}
$$

where "Intersect" operation finds the associated camera space point $(x_{u,v}^c, y_{u,v}^c, z_{u,v}^c)$ of $[u, v]$-indexed pixel on surface of 3D bounding box belonging to the oi-th object. Specifically, the oi-th object occludes other entities along the view ray from the pixel. Next, $\mathrm{proj}_{\mathrm{c2o}}^{\mathrm{oi}}$ transforms the coordinate from camera space to object space based on the pose of oi-th object. The "normalize" operation then maps the values to $[-1, 1]$ using the side lengths of 3D bounding box. Finally, we assign the feature calculated by the encoding function $f$ to $[u, v]$-indexed pixel in CNOCS map. There are many choices for $f$, leading to different variants such as: 1) Constant function. We can simply assign a predefined vector for any point, such as Euler angles. This variant is named C-CNOCS map. 2) Identity function. Similar to NOCS [52], we can directly use the coordinate $(x_{u,v}^{no}, y_{u,v}^{no}, z_{u,v}^{no})$ as point embedding. Unlike the original NOCS map, the points are located on surface of 3D bounding box instead of the precise shape determined by CAD model. This variant is named I-CNOCS map. 3) Spherical harmonic function. $(x_{u,v}^{no}, y_{u,v}^{no}, z_{u,v}^{no})$ can be further transformed into the form $(\theta_{u,v}^{no}, \phi_{u,v}^{no}, r_{u,v}^{no})$ in spherical coordinate system. Then, we can construct a series of Laplace's spherical harmonics $Y_l^m(\theta_{u,v}^{no}, \phi_{u,v}^{no})$ depending on a user-defined degree as point embedding, where $l, m$ represent indices of degree and order, respectively. This variant is named S-CNOCS map. Based on empirical results (Sec. 4), we use I-CNOCS map as the pose representation since it is simple and effective.

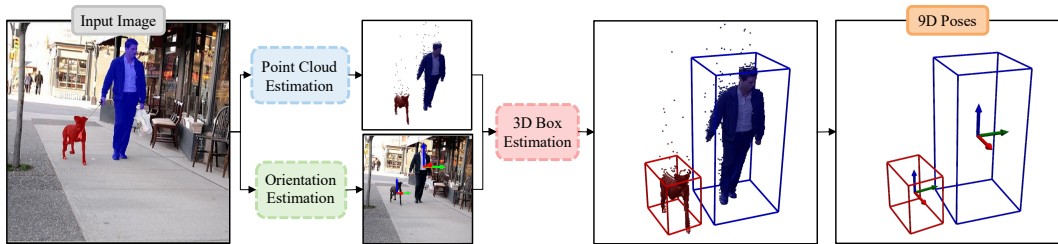

Figure 4: Annotation pipeline of 9D poses in MS-COCO [26].

### 3.4 ObjectPose9D Dataset

There are currently no existing datasets suitable for learning multi-object 9D pose control. Although some public datasets like Objectron [1] and OmniNOCS [20] contain object pose annotations, they are constrained by limited object categories and scene diversity. Therefore, we select several subsets from OmniNOCS as the base of our dataset, which cover common objects and scenarios in indoor and street scenes. Furthermore, we resort to the COCO dataset [26] to enlarge the data distribution and improve the model generalization ability. The annotation pipeline of 9D poses in COCO dataset [26] is shown in Fig. 4 and consists of the following steps:

1. **Selection of objects**. First, we obtain the suitable objects from each image using the following criteria: **1)** The area of object mask must be limited within predefined lower and upper bounds,

excluding the objects that are too small or large; **2)** The orientation of the objects should be unambiguous. Categories with inherent orientation ambiguity, such as "bottle", are excluded.

2. **Estimation of orientation and 3D bounding boxes**. Next, we obtain the orientations and 3D bounding boxes of suitable objects to construct CNOCS map. First, Orient Anything [59] is used to infer the object orientation and we filter the objects with low prediction confidence. In parallel, we get the initial geometry estimation (point clouds) of the scene via an advanced 3D reconstruction method [57], and identify the points belonging to the object based on the mask, while discarding the points that are too far from centroid by a threshold. Finally, 3D bounding boxes can be derived using the point clouds and predicted orientation, which tightly enclose the object points.

To ensure data quality, human annotators manually filter out low-quality samples and refine inaccurate annotations. Based on the estimated 3D bounding boxes and orientations, we then construct the CNOCS map representation using Eq. 2. Besides, Multimodal Large Language Model (MLLM) [5] is also employed to generate descriptive captions for each image, enriching the dataset with aligned textual information. These steps together yield the final dataset, *ObjectPose9D*. Further details on dataset statistics and construction procedures are provided in Appendix A.1.

### 3.5 SceneDesigner: Multi-object 9-DoF Pose Control

With the constructed dataset *ObjectPose9D*, we proceed to train our proposed method (*SceneDesigner*) for 9D pose control of multiple objects. The details about training and inference are introduced below.

**Learning for 9-DoF pose control:** We simply introduce ControlNet-like [65] branched network into the base model as overall architecture, which receives CNOCS map that encodes $\{\mathcal{P}_i\}_{i=1}^{N_o}$, text prompt $c_p$, and the noisy latent in the current step. $v_\theta$ is learned in two stages. In the first stage, the parameters from the branched network are optimized using Eq. 1, learning the basic 9D pose controllability. However, the learned model exhibits inferior performance on low-frequency poses from *ObjectPose9D*, which is caused by imbalanced pose distribution. For example, the model fails to generate the back views of most animals due to biases from datasets [20, 26]. Therefore, we resort to the technique from RLHF (Reinforcement Learning from Human Feedback). For aligning the object pose with the condition, we aim to maximize the introduced reward function below:

$$
\begin{aligned}
r_{ls}(x, c_p, \{\mathcal{P}_i\}_{i=1}^{N_o}) &= \Sigma_{i=1}^{N_o}\big(1 - D_{ls}(x, \mathrm{obj}_i, l_i, s_i)\big), \\
r_o(x, c_p, \{\mathcal{P}_i\}_{i=1}^{N_o}) &= \Sigma_{i=1}^{N_o}\big(1 - D_o(\mathcal{X}(x, \mathrm{obj}_i), o_i)\big), \\
r(x, c_p, \{\mathcal{P}_i\}_{i=1}^{N_o}) &= \gamma r_{ls}(x, c_p, \{\mathcal{P}_i\}_{i=1}^{N_o}) + \lambda r_o(x, c_p, \{\mathcal{P}_i\}_{i=1}^{N_o}),
\end{aligned}
\tag{3}
$$

where higher $r_{ls}$ represents better accuracy of location and size. For $r_{ls}$, we use the advanced detection model [29] to estimate the 2D bounding box, and compute its Intersection over Union (IoU) with the projected one from 3D bounding box. On the other hand, $r_o$ assesses the orientation precision. For $\mathcal{X}$, we crop the object area and feed it to Orient Anything [59]. Then, $D_o$ calculates the KL divergence between the target distribution and the estimated distribution from Orient Anything. The final reward $r$ is calculated by combining $r_{ls}, r_o$ through weighting factors $\gamma, \lambda$. Consequently, the objective of the second stage is to minimize:

$$
\mathbb{E}_{(c_p, \{\mathcal{P}_i\}_{i=1}^{N_o}) \sim \mathcal{B}, x \sim p_\theta(\cdot | c_p, \{\mathcal{P}_i\}_{i=1}^{N_o})} \left[ -\beta r(x, c_p, \{\mathcal{P}_i\}_{i=1}^{N_o}) + L_{prior} \right],
\tag{4}
$$

where text prompts and CNOCS maps are sampled from constructed balanced data distribution $\mathcal{B}$, and $L_{prior}$ is calculated as Eq. 1 for stabilizing the training [62]. However, naively training with Eq. (4) leads to huge GPU memory consumption for backpropagation throughout multi-step denoising process, which is infeasible. To reduce memory footprint, we leverage randomized truncated backpropagation and gradient checkpointing like in AlignProp [40], while feeding the coarse image estimated from intermediate step to the reward function instead of the clean one. The whole pipeline is presented in Algorithm 2. Through RL finetuning, the model outperforms in pose alignment with the input condition. More details about RL finetuning are provided in Appendix A.2.

**Inference:** After two-stage training, the model can effectively control the 9D pose of arbitrary object. However, the model fails to associate each object with its pose from CNOCS map in multi-object scenarios, since there is no restriction in constructing the correspondences. Furthermore, insufficient generation and attribute leakage often occur in generating multiple concepts. Therefore, we propose

**Algorithm 1:** Algorithm pipeline of *Disentangled Object Sampling*.

---

**Input:** Initial noise $\epsilon$ sampled from $\mathcal{N}(\mathbf{0}, \mathbf{I})$; text prompt $c_p$ consisting of entity names $\{obj_i\}_{i=1}^{N_o}$; CNOCS map for each object $\{\mathcal{P}_i\}_{i=1}^{N_o}$; CNOCS map of the whole scene $\mathcal{P}_{global}$; sampling step $T$;

$x_0 = \epsilon$;

**for** $t$ in $\{0, 1 \ldots, T-1\}$ **do**

    $v_t = v_\theta(x_t, t, c_p, \mathcal{P}_{total})$;

    $x_{t+1} = x_t + v_t dt$;

    **for** $i$ in $\{1, \ldots, N_o\}$ **do**

        Obtain object mask $M_i$ from CNOCS map $\mathcal{P}_i$;

        $v_t^i = v_\theta(x_t, t, obj_i, \mathcal{P}_i)$; /*it can be computed in parallel with $v_t$*/

        $x_{t+1}^i = x_t + v_t^i dt$;

        $x_{t+1} = (1 - M_i)x_{t+1} + M_i x_{t+1}^i$;

$x = x_T$;

**Return:** The generated image $x$.

---

*Disentangled Object Sampling* to alleviate the challenges, and the process is shown in Algorithm 1. Essentially, we combine multiple noisy latents at each denoising step using region masks, where each latent is sampled based on global or object-specific condition. Through *Disentangled Object Sampling*, the model is able to match each pose from CNOCS map with corresponding object. Furthermore, we can also load the personalized weights given by users and perform customized pose control of reference subjects.

# 4 Experiment

**Implementation details:** The proposed *SceneDesigner* is based on Stable Diffusion 3.5 [10], training with 6 NVIDIA A800 80G GPUs. We use AdamW optimizer with an initial learning rate of $5e^{-6}$. The resolution is set to $512 \times 512$ with 48 batch size. In the first stage, the parameters $\theta$ from the introduced ControlNet are updated for 45K iterations in the proposed *ObjectPose9D*. For the second stage, we further fine-tune the model with RL objective for 5K iterations. More details about training are provided in Appendix A.2. During inference, we only inject the conditions in initial 15 steps during sampling with 20 denoising steps.

**Validation details:** For validation metrics, we use mean Intersection over Union (mIoU) and spatial accuracy $Acc_{ls}$ for assessing the precision of location and size. Specifically, we use Grounding DINO [29] to detect generated objects. $Acc_{ls}$ is calculated as $\frac{\Sigma_{i=1}^{N} \mathcal{I}(\text{IoU}_i > 0.6)}{N}$, where $\mathcal{I}$ is indicator function and $N$ is total number of test cases. Similar to ORIGEN [32], we use the following two metrics for orientation evaluation:1) Abs.Err calculates the absolute error of azimuth angles (in degrees) between the input condition and the estimated one from Orient Anything [59] and 2)Acc.@22.5° measures the accuracy with 22.5° tolerance. Furthermore, CLIP [42] is used to estimate the text-image alignment and FID presents visual quality. Specifically, we randomly sample the reference images from LAION [46] to calculate FID. In addition, a user study is also conducted to complement the evaluation based on human preferences. For validation dataset, we introduce two benchmarks to assess the model performance in pose control of single-object and multi-object scenarios, named *ObjectPose-Single* and *ObjectPose-Multi*, which are obtained by estimating the 9D poses from validation part of COCO [26] as in Sec. 3.4. Among them, *ObjectPose-Single* is further divided into *ObjectPose-Single-Front* and *ObjectPose-Single-Back* for assessing the orientation accuracy in front- and back-facing scenarios, containing 247 and 156 samples, respectively. Besides, *ObjectPose-Multi* includes 229 cases.

## 4.1 Comparisons with State-of-the-Art Methods

**Evaluation in single-object generation:** This experiment evaluates the capability of single-object pose control. Although Zero-1-to-3 [28] exhibits considerable orientation controllability, it depends on the reference image from users and exhibits poor generalization in real-world images. Furthermore, since the codes of other relevant methods [32, 38] were not open-sourced at the time of our experiment, they are also not discussed in our experiment. Consequently, we choose LOOSECONTROL (LC) [6] and Continuous 3D Words (C3DW) [8] as compared T2I methods due to their abilities for 3D-

Table 1: Quantitative evaluation of pose alignment in multiple benchmarks.

| Benchmark | Method | Location&Size Alignment | | Orientation Alignment | |
|---|---|---|---|---|---|
| | | $Acc_{ls}$ (%)↑ | mIoU (%)↑ | Abs.Err↓ | Acc@22.5° (%)↑ |
| *ObjectPose-Single-Front* | C3DW [8] | 2.02 | 19.61 | 50.01 | 60.32 |
| | LOOSECONTROL [6] | 23.89 | 27.12 | 87.26 | 23.08 |
| | *SceneDesigner* (**ours**) | **50.20** | **57.21** | **13.23** | **89.47** |
| *ObjectPose-Single-Back* | LOOSECONTROL | 24.36 | 30.49 | 132.26 | 7.05 |
| | *SceneDesigner* (**ours**) | **52.56** | **60.66** | **17.47** | **83.33** |
| *ObjectPose-Multi* | LOOSECONTROL | 14.85 | 22.58 | 147.42 | 4.80 |
| | *SceneDesigner* (**ours**) | **47.16** | **52.16** | **23.14** | **80.79** |

Table 2: Comparisons of visual quality and text alignment.

| | C3DW [8] | LOOSECONTROL [6] | *SceneDesigner* (**Ours**) |
|---|---|---|---|
| FID↓ | 67.39 | 37.89 | **24.91** |
| CLIP↑ | 0.267 | 0.293 | **0.345** |

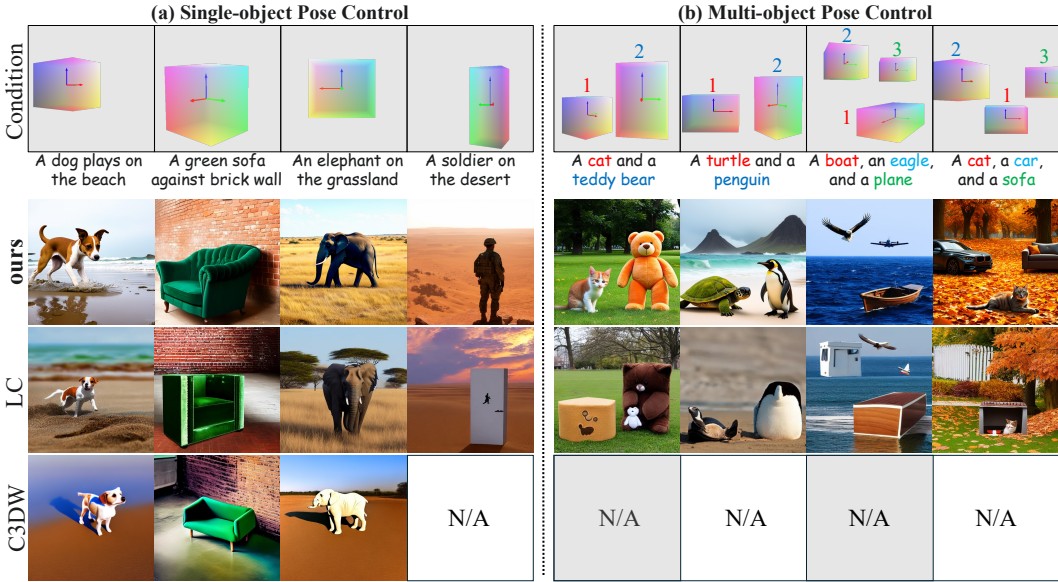

Figure 5: Evaluation of 9D pose control in single- and multi-object scenarios. *SceneDesigner* outperforms other methods in both fidelity and quality under various pose conditions.

aware control, while converting the pose condition into their required formats. As shown in the left of Fig. 5, the compared methods exhibit limited ability to control the location, size, and orientation simultaneously. Although LC [6] can control the location and size of the object effectively, the results present arbitrary orientation with poor quality and fidelity. On the other hand, C3DW is suffered by invariant image layout and unrealistic visual content. Besides, it can only process the front 180° range of azimuths. As a result, the method lacks the controllability in more complicated orientations (*e.g.*, back-facing case in column 4 of Fig. 5), and is unable to locate the object with a designated size. In contrast, *SceneDesigner* outperforms in both fidelity and quality. The quantitative results in Tab. 1 also demonstrate the superior performance of our method in pose alignment. As demonstrated by $Acc_{ls}$ and mIoU metrics in Tab. 1, *SceneDesigner* outperforms LC in controlling spatial location and size of the object, while C3DW exhibits poor precision since the objects are typically centered in the image. For orientation, C3DW achieves higher performance than LC, since the latter does not encode the orientation properties. However it cannot generate back-facing object, and suffers from the low-quality contents as indicated by the FID metric in Tab. 2. On the other hand, our method achieves considerable performances in both front- and back-facing settings with the help of efficient representation and the two-stage training strategy, while outperforming in both quality and text alignment.

**Evaluation in multi-object generation:** We also consider the pose control of multiple objects in the same scene. Since C3DW can only handle single object generation, LC is chosen as the compared method in this setting. As shown in the right of Fig. 5, it demonstrates poor performance in both

Table 3: Ablation studies.

| Benchmark | Setting | Location&Size Alignment | | Orientation Alignment | |
|---|---|---|---|---|---|
| | | Acc$_{ls}$ (%)↑ | mIoU (%)↑ | Abs.Err ↓ | Acc@22.5° (%) ↑ |
| *ObjectPose-Single* | w/o MS-COCO [26] | 41.69 | 50.07 | 74.89 | 24.32 |
| | w/o RL finetuning | 43.18 | 50.32 | 43.85 | 52.36 |
| | w/ C-CNOCS map | 40.45 | 49.86 | 37.86 | 73.70 |
| | w/ Pose embedding | 32.51 | 40.73 | 49.65 | 47.15 |
| | prompt only | 12.90 | 14.32 | 88.43 | 25.31 |
| | *SceneDesigner* (**ours**) | **51.12** | **58.55** | **14.87** | **87.10** |
| *ObjectPose-Multi* | w/o *DOS* | 36.68 | 45.91 | 42.92 | 59.39 |
| | *SceneDesigner* (**ours**) | **47.16** | **52.16** | **23.14** | **80.79** |

Table 4: Human preferences in quality, pose fidelity, and text-to-image alignment.

| Method | C3DW [8] | LOOSECONTROL [6] | *SceneDesigner* |
|---|---|---|---|
| Image quality | 0.47 | 0.64 | **0.96** |
| Location fidelity | 0.05 | 0.88 | **0.98** |
| Size fidelity | 0.02 | 0.82 | **0.96** |
| Orientation fidelity | 0.62 | 0.39 | **0.91** |
| Text-to-image alignment | 0.63 | 0.51 | **0.94** |

orientation and semantic fidelity, while suffering from insufficient generation and concept confusion. As a result, the method cannot precisely identify the concept inside each 3D bounding box. In comparison, *SceneDesigner* leverages *Disentangled Object Sampling* during inference to mitigate the entangled generation, while maintaining the pose controllability. The quantitative results from Tab. 1 also demonstrate that *SceneDesigner* maintains performance comparable to single-object scenarios when handling multiple objects, outperforming other methods in all metrics.

**User Study:** We additionally conduct a user study to assess the methods in the following aspects: *image quality*, *pose fidelity* and *text-to-image alignment*. Tab. 4 demonstrates the evaluation results from human beings, where the scores are normalized and the higher value indicates better performance. Specifically, we employed 20 volunteers to evaluate outcomes from each method. For each score, we average the results and normalize it by the maximum value.

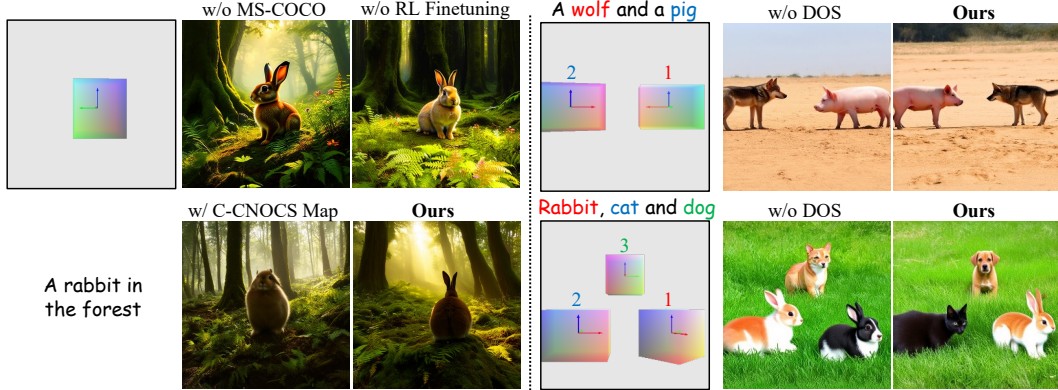

Figure 6: Ablation studies. The left indicates the effects of *ObjectPose9D*, two-stage training strategy and I-CNOCS map. The right examples show the impact of *Disentangled Object Sampling* (*DOS*).

## 4.2 Ablation Studies

We conduct comprehensive ablation studies to validate the effectiveness of each component in our proposed *SceneDesigner*. The quantitative results are summarized in Tab. 3.

**Pose Conditioning:** To validate our proposed CNOCS map, we compare it with three alternative conditioning strategies: 1) a C-CNOCS map that assigns constant Euler angles to the object region; 2) a variant that directly injects 9D pose embeddings via attention; and 3) a training-free baseline (*prompt only*) that converts the pose into a textual description based on templates. As shown in Tab. 3, all three baselines underperform, confirming the superiority of our representation.

**Dataset and Training Strategy:** We then analyze the impact of our dataset and two-stage training strategy. First, we train a model using only OmniNOCS data. As shown in Fig. 6 (left) and Tab. 3, the limited categories in OmniNOCS lead to poor generalization on unseen classes (e.g., rabbit) and a significant performance drop. Second, we evaluate the model checkpoint from the first stage, prior to RL finetuning. This model fails to generate back-facing objects, and its orientation accuracy is considerably lower. These comparisons confirm the importance of both our diverse *ObjectPose9D* and the RL finetuning stage.

**Multi-Object Generation with *DOS*:** For multi-object generation scenarios, we assess the effect of the proposed *DOS*. The results in the right part of Fig. 6 and Tab. 3 highlight its effectiveness in mitigating concept confusion and ensuring each object correctly corresponds to its specified pose in the CNOCS map.

## 5 Limitations and Impacts

### 5.1 Limitations

Although *SceneDesigner* achieves high-fidelity pose control, it cannot control the precise shape of the object. Furthermore, the performance in the multi-object scenario is constrained by the inherent capability of the base model. As mentioned in previous literature, increasing the number of semantic concepts in text prompt exacerbates insufficient generation and attribute leakage. While our *Disentangled Object Sampling* technique mitigates this issue, it introduces additional computation. Therefore, we will explore how to enhance the alignment with conditions in multi-object generation while maintaining computational efficiency in our future work.

### 5.2 Social Impacts

**Positive societal impacts:** *SceneDesigner* supports users with ability in multi-object 9D pose control without extensive resources or professional equipment, while providing customized pose control of user-provided subjects. This capability proves particularly valuable for applications like virtual/ augmented reality and product design, where spatial control is essential.

**Potential negative societal impacts:** Multi-object 9D pose control raises concerns about potential misuse for generating deceptive content in sensitive areas such as political manipulation and social media, where inaccurate poses could spread misinformation. Without proper safeguards, such as detection methods, ethical guidelines, and public awareness, this technique could be exploited to undermine the trust in digital media.

**Mitigation strategies:** Developing and adhering to strict ethical guidelines for multi-object 9D pose control technologies helps mitigate misuse risks. This includes enforcing usage restrictions for sensitive generation and ensuring transparency in generated outputs through traceability measures.

## 6 Conclusion

We introduce *SceneDesigner* that achieves multi-object 9D pose control within the same scene. Our key insight is introducing CNOCS map to encode the 9D pose of general objects, preserving the 3D properties and exhibiting high geometry interpretation. This representation accelerates the convergence and facilitates to provide a user-friendly interface for creating 3D-aware conditions. For learning the newly added control modules, we resort to the public dataset with comprehensive 9D pose annotations, while enhancing the diversity through annotating large-scale datasets. To alleviate the poor performance in low-frequency poses caused by imbalanced data distribution, we introduce a two-stage training process, where the second stage maximizes the proposed reward function to enhance the alignment of object pose with input conditions. During inference, our *Disentangled Object Sampling* associates each object with the corresponding pose condition, avoiding the confusion in multi-object generation. Furthermore, our method can also achieve customized pose control of reference subjects given by users. Finally, qualitative and quantitative experiments demonstrate that our method outperforms existing methods in both single- and multi-object scenarios.

**Acknowledgement.** This project was supported by the National Natural Science Foundation of China (NSFC) under Grant No. 62472104. This work was supported by Xiaomi Young Scholars Program.

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

# A  Technical Appendices and Supplementary Material

**Overview:** The supplementary includes the following sections:

- **A.1.** Details of dataset.
- **A.2.** Details of method.
- **A.3.** More comparisons and analyses.
- **A.4.** More ablation studies.

## A.1  Details of Dataset

As the base of our dataset *ObjectPose9D*, we select Objectron [1] and Cityscapes [9] subsets from OmniNOCS [20], sampling around 110,000 images with comprehensive pose annotations of interesting instances. Furthermore, we enlarge the category diversity and scene variations through introducing additional data from MS-COCO [26], and leverage the approach in Sec. 3.4 to obtain pose annotations for suitable objects, obtaining around 65,000 samples. Concretely, we select objects whose sizes range from 10% to 70% of the image size, and further exclude those with prediction confidence [59] below 0.8. For estimation of 3D bounding boxes, the farthest 10% of point clouds from the object centroid are discarded. In addition, Qwen2.5-VL-7B [5] is used to describe the visual content. To maintain high-quality training data, human annotators further filter the low-quality images and rectify the inaccurate annotation. Finally, *ObjectPose9D* contains totally 125,486 training data.

## A.2  Details of Method

**Differences with related methods:** There exists significant differences between our method and related approaches. **1) Controllability of 9D poses.** To the best of our knowledge, *SceneDesigner* is the first method that can achieve multi-object 9D pose control. For example, LOOSECONTROL [6] that receives 3D bounding boxes is unaware of object orientation, and the results are significantly influenced by model priors. Other methods [38, 8] like Continuous 3D Words [8](C3DW) can only control the front $180°$ azimuth angles, and lack flexibility in scene composition, typically producing rigid arrangements. The recent work ORIGEN [32] is able to achieve intricate orientation control, but limited in controlling the location and size. **2) Dataset and training strategy.** The high quality training samples and efficient learning strategy make *SceneDesigner* outperform other methods in terms of quality and fidelity. LOOSECONTROL with single-stage learning performs poor alignment with conditions in some complicated scenarios. Methods [38, 8] trained on synthetic dataset exhibit inferior generalization ability in generating real-world images, thus having poor image quality. The training-free method ORIGEN optimizes the initial noise to maximize the introduced reward function, aligning the object orientation with input condition. However, it is only applicable to one-step generative models, hindering its application.

**Details about RL finetuning:** Constructing a balanced pose distribution for each object category from existing data sources [26, 46] is impractical. Therefore, RL finetuning discussed in Sec. 3.5 is introduced to alleviate the challenge. In our implementation, we set $\gamma$ and $\lambda$ in Eq. (3) to 0 and 1, respectively, since our experiment shows that $L_{prior}$ effectively maintains control over both location and size. $\beta$ in Eq. (4) is set to $5e^{-3}$ to avoid overfitting. The dataset $\mathcal{B}$ from Eq. (4) constructs balanced pose distribution for each considered object category. An ideal approach is to randomly place multiple cuboids of arbitrary sizes and orientations in the space, while generating corresponding CNOCS maps. However, our experiments demonstrate that this leads to slow convergence. Consequently, we only consider single-object scenarios. In practice, the model trained under this setting demonstrates strong generalization capability in multi-object scenarios through *Disentangled Object Sampling* technique. We get totally 20,000 CNOCS maps with different location, size, and orientation. Besides, we create 1200 descriptions of interesting object categories (*e.g.* animal) by querying MLLM [5]. Then, we can independently sample the text prompt and CNOCS map to construct balanced pose distributions for rich object categories. The pipeline of RL finetuning is illustrated in Algorithm 2. We optimize the whole parameters from ControlNet with truncation length $K$ set to 2. The denoising steps are sampled from uniform distribution $U(T_{min}, T_{max})$, where $T_{min}, T_{max}$ are set to 6 and 16.

## A.3  More Comparisons and Analyses

**More qualitative results:** To illustrate the flexibility of our method in 9D pose control, Fig. 7 presents the results under various pose conditions. As shown in the figures, the poses of the generated objects exhibit strong fidelity with conditions. Besides, Fig. 8 and Fig. 9 present more qualitative comparisons in single- and multi-object scenarios with intricate pose conditions. These results also demonstrate that *SceneDesigner* outperforms other methods in terms of fidelity and image quality. Fig. 10 further presents more qualitative results of our method in multi-object generation. We also provide examples in Fig. 11 to present the model's capability of customized pose control. Specifically, we use images from DreamBooth [45] and learn LoRA [15] weights for each subject.

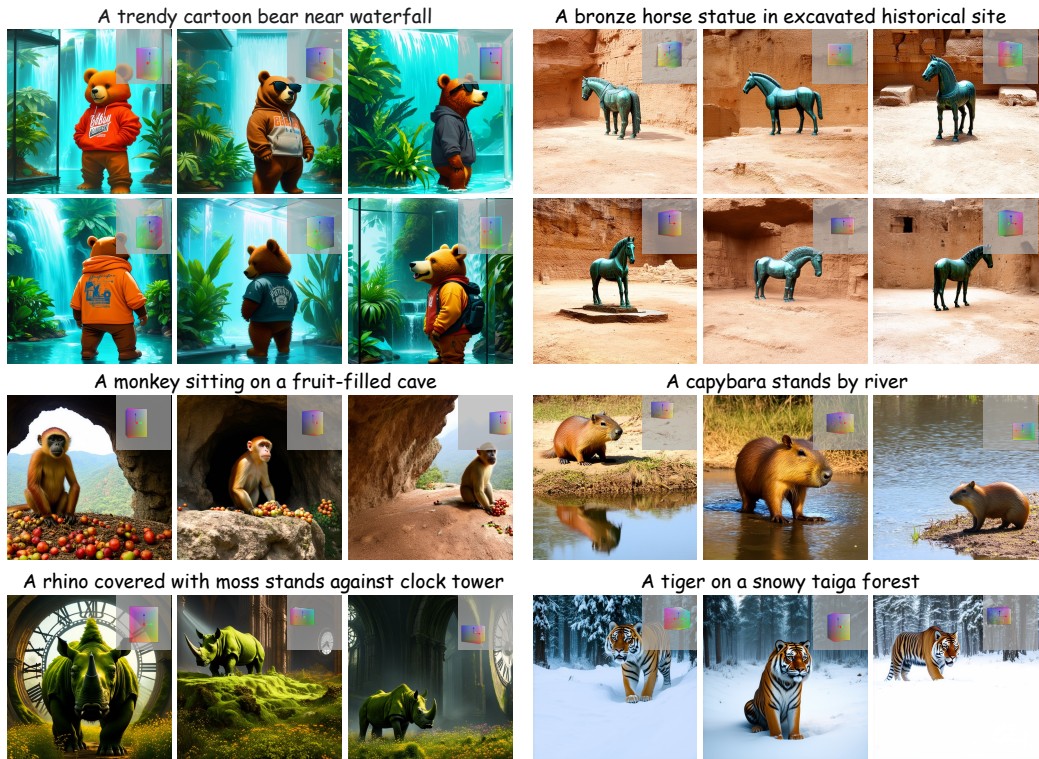

Figure 7: 9D pose control under various pose conditions.

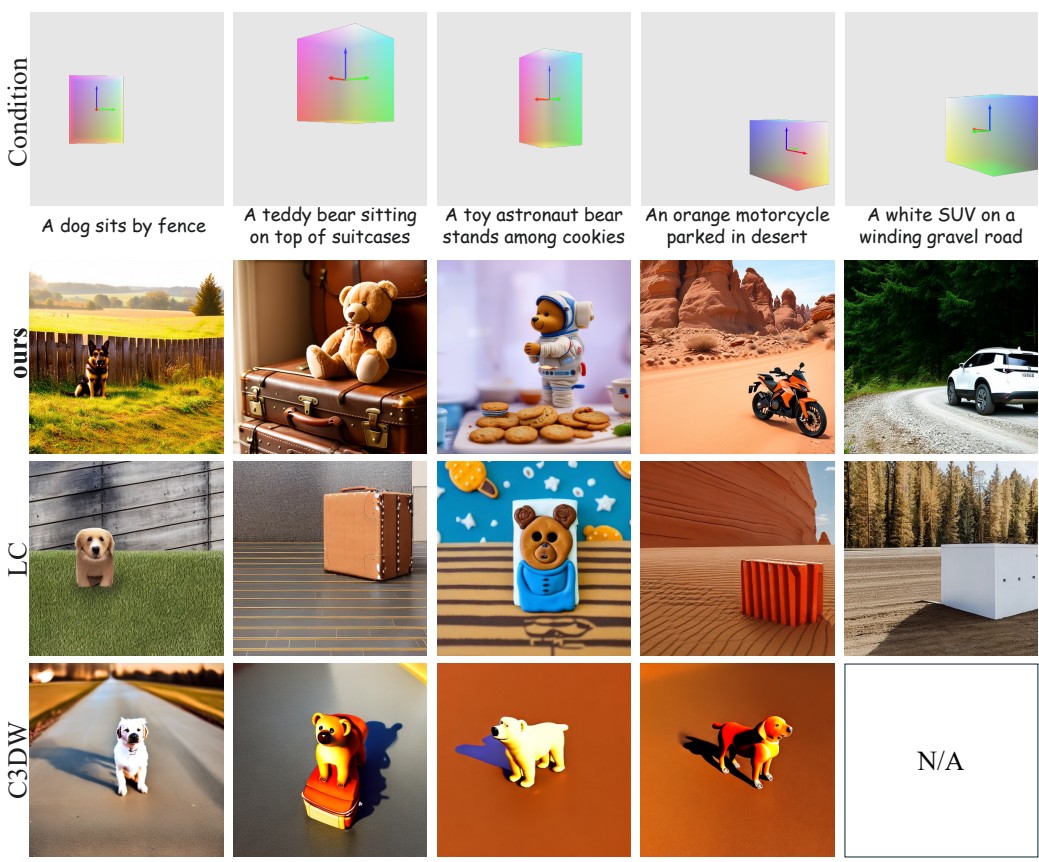

Figure 8: Evaluation of 9D pose control in single-object scenarios.

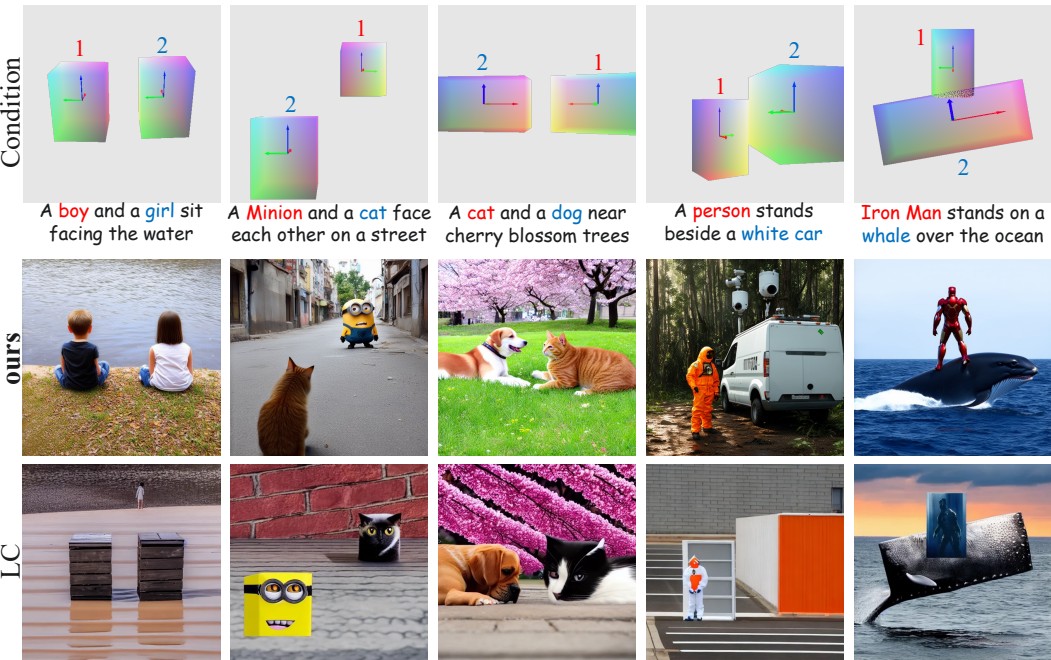

Figure 9: Evaluation of 9D pose control in multi-object scenarios.

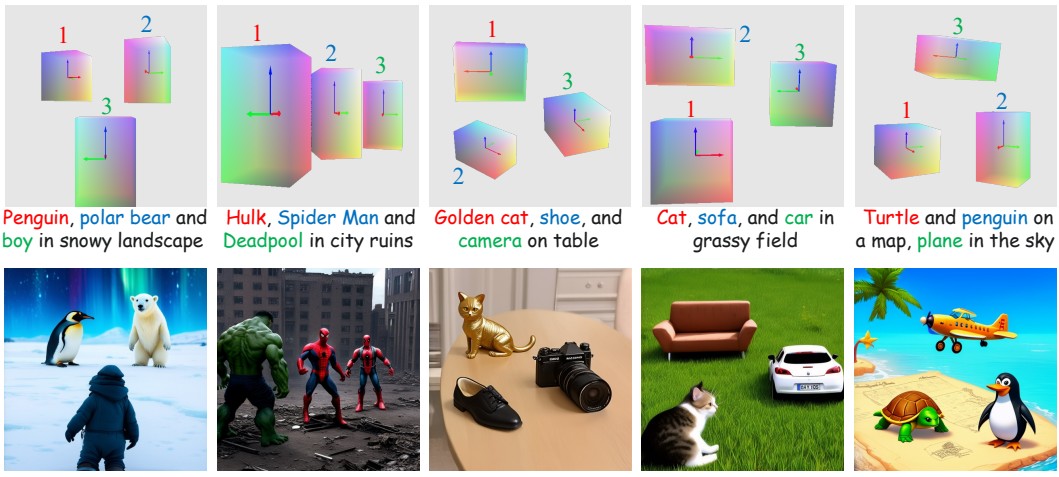

Figure 10: More generation results of *SceneDesigner* in multi-object scenarios.

**Algorithm 2:** Algorithm pipeline of RL finetuning.

---

**Input:** The proposed dataset *ObjectPose9D* $\mathcal{D}$ and constructed dataset $\mathcal{B}$ for RL finetuning; truncation length $K$; the range of denoising steps $[T_{\min}, T_{\max}]$ during sampling; the number of training epochs $N_E$; weighting factor $\beta$ of reward function;

**for** *epoch* in $\{1\ldots, N_E\}$ **do**

  Get samples from $\mathcal{D}$ and update the network parameters $\theta$ through Eq. (1);

  Sample the CNOCS map that encodes $\{\mathcal{P}_i\}_{i=1}^{N_o}$, and $c_p$ from $\mathcal{B}$;

  Obtain the initial noise $\epsilon$ sampled from $\mathcal{N}(\mathbf{0}, \mathbf{I})$;

  Obtain the sampling steps $T_1$ from uniform distribution $U(T_{\min}, T_{\max})$;

  Sample the step $T_0$ from uniform distribution $U(T_1 - K, T_1 - 1)$, which begins gradient calculation;

  $x_0 = \epsilon$;

  **for** $t$ in $\{0, \ldots, T_0 - 1\}$ **do**

    no grad : $x_{t+1} = x_t + v_\theta(x_t, t, c_p, \{\mathcal{P}_i\}_{i=1}^{N_o})dt$;

  **for** $t$ in $\{T_0, \ldots, T_1 - 1\}$ **do**

    with grad : $x_{t+1} = x_t + v_\theta(x_t, t, c_p, \{\mathcal{P}_i\}_{i=1}^{N_o})dt$;

  $\hat{x} = \frac{x_{T_1} - T_1\epsilon}{1 - T_1}$;

  Calculate the gradient towards $-\beta r(\hat{x}, c_p, \{\mathcal{P}_i\}_{i=1}^{N_o})$ defined in Eq. (3) and update the $\theta$;

**Return:** The network parameters $\theta$.

---

Table 5: Ablation studies.

| Setting | Location&Size Alignment | | Orientation Alignment | |
|---|---|---|---|---|
| | $\text{Acc}_{ls}$ (%)↑ | mIoU (%)↑ | Abs.Err ↓ | Acc@22.5° (%) ↑ |
| w/o $L_{prior}$ | 26.55 | 31.76 | 28.79 | 76.18 |
| *SceneDesigner* (ours) | **51.12** | **58.55** | **14.87** | **87.10** |

## A.4   More Ablation Studies

We conduct additional experiments to demonstrate the impacts of critical techniques from our method. Beyond the settings discussed in the main paper, we further validate the effectiveness of $L_{prior}$ loss in Eq. (4). Tab. 5 and qualitative results in Fig. 12 indicate that the model learned without $L_{prior}$ suffers from reward hacking and quality degradation under the same training configuration. Furthermore, Fig. 13 shows the impacts of our *Disentangled Object Sampling* (*DOS*).

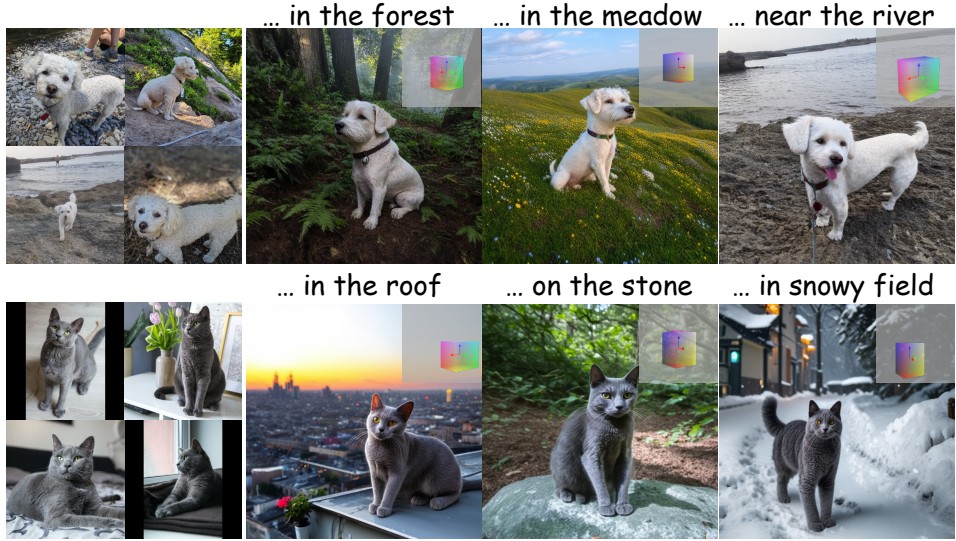

Figure 11: Customized 9D pose control.

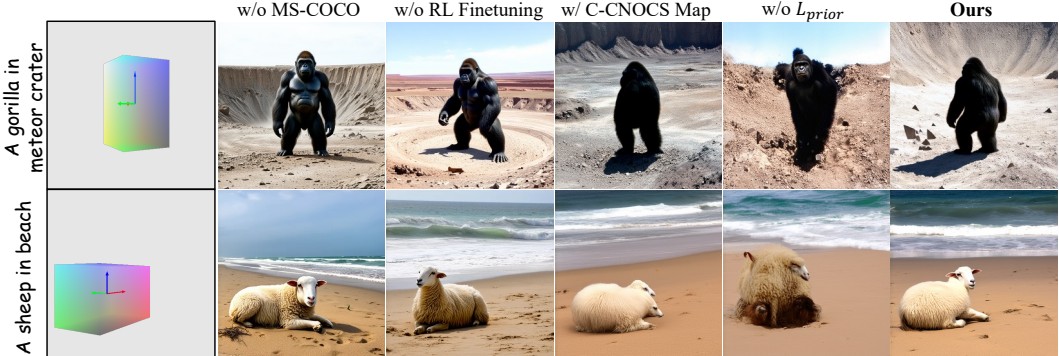

Figure 12: Ablation study in *ObjectPose9D*, two-stage training strategy and I-CNOCS map. The model learned without $L_{prior}$ suffers from overfitting and quality degradation.

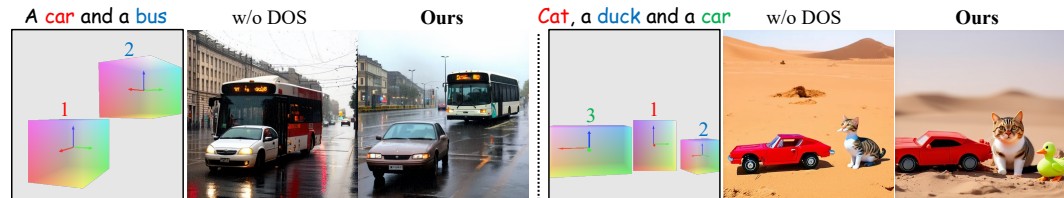

Figure 13: Ablation study in *Disentangled Object Sampling* (*DOS*). Our method alleviates the insufficient generation and concept fusion with the help of *DOS*.

