# OpenReview forum: "SceneDesigner: Controllable Multi-Object Image Generation with 9-DoF Pose Manipulation"
_NeurIPS.cc/2025/Conference — NeurIPS 2025 spotlight_

### Official Review · Reviewer_erLh · 2025-06-03

**Clarity:** 3
**Significance:** 3
**Originality:** 3
**Rating:** 5
**Confidence:** 4

**Summary:**

This paper presents SceneDesigner, a controllable image generator that can control the 3D pose of objects. SceneDesigner builds upon a pre-trained flow matching model, and leverages the ControlNet-style conditioning mechanism. Specifically, it converts object 3D poses into a CNOCS map which has the same shape as the generated image. To train the model, the authors curate ObjectPose9D, a large-scale dataset with object 3D pose annotations. The dataset consists of existing datasets with limited object categories and automatically labeled images from COCO. To improve the model's performance in rare poses (e.g. back view), the authors further apply REFL-style reward optimization. Experimental results demonstrate a notable improvement over baselines, both quantitatively and qualitatively.

**Questions:**

Besides the above weaknesses, I have the following questions:
1. Lines 151-152 claim that directly projecting pose conditioning to embeddings and injecting to networks via attention "leads to slow convergence as indicated by experiments in previous works [49, 21, 22]." I do not think any of the cited papers discussed this. Thus, it is improper to cite them here. This is related to the second weakness above. Prior works such as Compass Control and Neural Assets seem to suggest that 3D controllability can be achieved with direct projection and fine-tuning on only a few thousand images. I feel it should be even easier here as the proposed ObjectPose9D dataset is much larger. Can the authors conduct an ablation here?
2. I'm surprised that the reward optimization stage takes 5k steps. From my experience, REFL-style training leads to reward hacking (over-optimization) very quickly after 1-2k steps. Did the authors also observe similar effects in their experiments if trained longer?
3. I agree that the proposed DOS can effectively associate objects to their 3D bboxes. However, wouldn't using hard masking result in artifacts like unsmooth transitions around object boundaries? What about using something like MultiDiffusion [1]?
4. There seems to be a "front-facing" issue in the generated images, e.g., the bear in row 2 column 2 of Fig. 8 and the sheep in Fig. 12 last column -- the object tries to turn its head to "look at" the camera. Can authors discuss why this issue happens? How common is it in the generation results?

[1] Bar-Tal, Omer, et al. "MultiDiffusion: Fusing Diffusion Paths for Controlled Image Generation." ICML. 2023.

**Ethical Concerns:**

["NO or VERY MINOR ethics concerns only"]

**Final Justification:**

The authors rebuttal addressed all of my major concerns. I believe the works is a simple and scalable approach in the field of 3D pose controllable generation. In addition, the author promised to release the dataset. As far as I know, there is no such large-scale pose-annotated in-the-wild dataset available in the literature. I believe it is also a great contribution to the field. Thus, I recommend Acceptance of the paper.

**Limitations:**

- What will happen if we want to generate overlapping objects? Will DOS still work? Fig. 10 already has some examples with small object intersections, but I am curious about more heavily occluded objects.

**Paper Formatting Concerns:**

N.A

**Quality:**

3

**Strengths And Weaknesses:**

## Strength

1. Lack of real-world training data is a long-standing issue in training 3D-aware controllable generators. The automatic labeling pipeline on in-the-wild COCO dataset is a scalable method that can be adapted to more images. The ablation study also shows the importance of this additional data source;
2. As far as I know, the use of NOCS map as 3D object pose representation in controllable generation is novel. It can represent full 360 degree object orientations which is lacking in prior work such as LooseControl;
3. The reward optimization is also reasonable and effective. It helps improve long-tail pose conditioning as we do not need GT images with these poses, but only pose detectors as verifiers.

## Weakness

1. A few papers you may want to discuss in Related Works. [1] is a 3D-aware image editing method following a similar setting as 3DIT. [2] is 3D-aware video model that seems to work well in multi-object and in-the-wild cases. Comparing with these methods is not required as the settings are different;
2. The effectiveness of the CNOCS map is not really demonstrated. While there are ablation studies on C-CNOCS vs I-CNOCS, I would like to see comparison of CNOCS map vs directly projecting 3D poses (e.g. the one Compass Control, [1], and [2] are using);
3. While Disentangled Object Sampling (DOS) is used at inference time to associate objects with its 3D pose conditioning, there is no such association in the training stage. In my opinion, this inevitably leads to entanglement between different objects. Why not use something like CALL in Compass Control?
4. (Minor) typos and writing issues.
- Lines 38, 230, 245: "it's" --> "its";
- Lines 287, 298, 315, "in (the) other hand" --> "on the other hand";
- Eq. 2: maybe change "oi" to "j" as it's more frequently used as subscript?

[1] Wu, Ziyi, et al. "Neural assets: 3d-aware multi-object scene synthesis with image diffusion models." NeurIPS. 2024.

[2] Fu, Xiao, et al. "3DTrajMaster: Mastering 3D Trajectory for Multi-Entity Motion in Video Generation." ICLR. 2025.

---

> ### Author Rebuttal · Authors · 2025-07-31
>
> ## W1 More discussions about relevant papers
> Thank you for highlighting these relevant papers. We appreciate the suggestions and will incorporate a discussion of these works in the "Related Works" section of our revised paper.
>
> ---
>
> ## W2 The effectiveness of the CNOCS map
> We sincerely thank you for this valuable suggestion. To better demonstrate the effectiveness of our proposed CNOCS map, we additionally trained a model that directly receives encoded 9D pose embeddings of objects, which are injected into the network via the attention mechanism. We evaluated this model on the ObjectPose-Single benchmarks ("Pose embedding (45k)" in the "Setting" column).
>
> | Setting              | $Acc_{ls}$ (%)↑ | mIoU (%)↑ | Abs.Err ↓ | $Acc$@22.5° (%)↑ |
> |----------------------|-------------------|-----------|-----------|----------------|
> | Pose embedding (45k) | 32.51             | 40.73     | 49.65     | 47.15          |
> | **SceneDesigner** (ours, 20k)      | 39.21             | 46.92     | 52.19     | 46.40          |
> | **SceneDesigner** (ours, 45k)       | 43.18             | 50.32     | 43.85     | 52.36          |
>
> We conducted a comparison of models under 45k training iterations (rows 1 and 3), as well as our model trained for 20k steps (row 2).
>
> From the comparison between the first two rows, our method achieves performance comparable to the pose embedding-based model (trained for 45k steps) after only 20k training iterations (with higher accuracy in location&size and slightly lower accuracy in orientation). This demonstrates that the CNOCS map facilitates faster convergence. Under the same training configuration (row 1 and row 3), the model using pose embedding exhibits inferior performance on the benchmark. This further indicates that the CNOCS map helps the model achieve high-fidelity pose control compared to directly projecting 9D poses.
>
> Below, we analyze the reasons for this performance gap. 1. Unlike directly projecting 9D poses, the CNOCS map preserves spatial information by rendering 3D bounding boxes into the image space. Thus, our method outperforms in representing the location and size. 2. The CNOCS map provides dense correspondences between image pixels and their normalized 3D coordinates in the object space. Compared to the global guidance from projected 9D poses, this **pixel-wise guidance** offers stronger geometric interpretability, facilitating the network to capture the relationship between the object appearance and 3D structure, thereby better encoding the orientation information.
>
> In summary, both the experimental results and analysis demonstrate that the CNOCS map outperforms projected 9D poses in representing object poses.
>
> ---
>
> ## W3  The object entanglement problem
> Thank you for your valuable suggestion! We agree that incorporating additional supervision during training is an effective strategy for resolving object entanglement. Indeed, without such supervision and DOS, entanglement issues arise, as shown in our ablation study (Figures 6,13, w/o DOS). Our DOS algorithm effectively addresses this at inference time. Benefiting from the high reliability of our single-object pose control, DOS achieves accurate multi-object control by independently sampling each object region and then integrating them. The effectiveness of this approach is also demonstrated by the quantitative results in Table 3 of our main paper.
>
> Nevertheless, we acknowledge the high effectiveness of your suggestion and plan to integrate training-stage supervision in our future work, as this indeed reduces the computational overhead during inference.
>
> ---
>
> ## W4 Typos and writing issues
> Thank you for your valuable suggestions! We will correct typos and writing issues in the revised version.
>
> ---
>
> ## Q1(a) Improper citations
>
> We sincerely apologize for improper statements and citations, and commit to removing this part in the revised paper.
>
> Our initial intention was to highlight their reported computational cost of training on large-scale datasets. For instance, GLIGEN [22] utilized 16 V100 GPUs to learn the model for 100k iterations (64 batch size) on COCO2014. Furthermore, as indicated by our response to "Weakness 2", the CNOCS map exhibits better convergence speed and generalization ability compared to projected 9D pose under the same training configuration.
>
> Nevertheless, we recognize that it is improper to declare "this approach leads to slow convergence as indicated by experiments in previous works [49,21,22]" in the "Method" section. We apologize again for this improper citation and will remove the statements from the revised version.
>
> ---
>
> ## Q1(b) Why Compass Control and Neural Assets only need a few images
>
> Thank you for your valuable question! In following, we list training settings of our model and the mentioned methods based on direct pose projection.
>
> * **Neural Assets**: The training data of Neural Assets are filtered from 400k images and 26k videos. The model was trained for 250K iterations on 256 TPUs.
>
> * **Compass Control [36]**: The training data of Compass Control consists of 15k images. The model was trained on a single A6000 GPU for 25k iterations.
>
> * **Our Method**: The training data of our method consists of 125k images. The model was trained on 6 A800 GPUs for 45k iterations in the stage 1.
>
> Neural Assets requires more data and training steps compared to our method. Compass Control, trained on a small synthetic dataset (15k, images are rendered by predefined rules), achieves fast convergence due to the limited object categories and single-style visual content. However, such models relying on synthetic datasets struggle to generate realistic images due to domain gap, as evidenced by the results of Continuous 3D Words (C3DW) [8] in Figures 5,9 of our main paper.
>
> To achieve precise 9-DoF pose control across diverse object categories and generate photorealistic images, our model was trained on a large-scale real-world dataset (125k images) with a broader range of the scene variations. Learning the appearances of diverse object categories with various poses requires a large amount of training data as support. Consequently, training with limited data distribution will degrade the model performance, as demonstrated in the Figures 6,12 and Table 3 (w/o MS-COCO) from our main paper.
>
> Furthermore, our response to "Weakness 2" highlights that under the same training configuration, the CNOCS map representation outperforms direct pose projection in both convergence speed and generalization ability. This controlled experiment demonstrates the effectiveness of CNOCS maps for 9-DoF pose control.
>
> ---
>
> ## Q2 Reward hacking (over-optimization) problem
>
> You are right. When the training objective is solely to improve the reward function, "reward hacking" (over-optimization) occurs very quickly—typically between 1k and 2k steps, as you mentioned.
>
> To avoid rapid overfitting and stabilize the fine-tuning process, we regularize the training using the pre-training loss $L_{prior}$ (calculated as Equation 1 from our main paper). Lines 557-559 in the appendix and the 5th column (w/o $L_{prior}$) of Figure 12 demonstrate that training for 5k steps without $L_{prior}$ leads to quality degradation, whereas our method generates high-quality and high-fidelity images.
>
> However, even with $L_{prior}$, the "reward hacking" problem still occurs when the model is trained for 6k to 7k steps. We plan to further optimize the algorithm in our future research.
>
> ---
>
> ## Q3 Suggestions for the proposed DOS
>
> We sincerely thank you for this valuable suggestion. To be frank, most of our generated images exhibit natural visual content without artifacts like unsmooth transitions around object boundaries. As shown in its official code, MultiDiffusion is equivalent to DOS when there is only one object (the noisy latents of the object and background are aggregated according to the mask). When there are multiple objects, MultiDiffusion takes a weighted average of the latents derived from each object in the overlapping area. We leveraged this idea and found that it improves the image quality in some test cases, where the overlapping area is within a certain range (40-60% of object area). We will update the DOS algorithm in our revised version to incorporate this strategy, and will cite MultiDiffusion in our paper.
>
> ---
>
> ## Q4 "Front-facing" issue
> Thank you for this insightful observation. We believe this phenomenon stems from the bias of MS-COCO part from our dataset. Our investigation reveals that some animals with left/right/backward-facing bodies sometimes have their faces turned toward the camera, likely due to photographers' creative intentions. However, the estimation model (Orient Anything) primarily uses body orientation as its prediction. These lead our model to occasionally generate "front-facing" animals with body orientations that conform to the input conditions.
>
> According to our investigation, the orientation of the animal's head is consistent with its body in most cases.
>
> ---
>
> ## Limitations: What will happen if overlapping objects are generated
> We have conducted additional experiments for scenarios with significant occlusion. We found that the performance of the model deteriorates as the overlapping area increases. Specifically, when 50-70% of the object is occluded, the model occasionally generates the truncated body. when 70-100% of the area is occluded, the object is generated in incorrect location, significantly compromising pose fidelity. Thanks for your insightful feedback. We will add this limitation to our revised version, and will explore to enhance the model performance in this challenging scenario.

---

> ### Comment · Reviewer_erLh · 2025-08-01
> **Re: Rebuttal**
>
> I thank the authors for their efforts in preparing a strong rebuttal. My main concerns are all addressed. I have one additional question: will the ObjectPose9D dataset be released in the future? I believe it will be a huge contribution to the controllable generation community. I understand there might be concerns about data release so the answer will not affect my rating, as I think the contribution of this work is enough for acceptance.

---

> > ### Author Response · Authors · 2025-08-01
> > **Commitment to Release the ObjectPose9D Dataset**
> >
> > Dear Reviewer erLh,
> >
> > Thank you very much for your positive feedback and for recognizing the value of our work. We sincerely appreciate your thoughtful review and are glad to hear that your main concerns have been addressed.
> >
> > Regarding your question: **yes**, we confirm that the **ObjectPose9D dataset will be released** upon acceptance. We believe this will serve as a valuable resource for the community, especially in advancing controllable generation research.
> >
> > If this clarification helps resolve your remaining question, we would be very grateful if you could consider reflecting it in your final rating. Thank you again for your time and constructive comments.
> >
> >
> > Best regards,
> > Authors of Paper #2041

---

> > > ### Comment · Reviewer_erLh · 2025-08-02
> > > **Re Dataset Release**
> > >
> > > Thank you for confirming this. I am happy to raise my score. Please incorporate additional experimental results, discussions of related works and limitations into the final version of the paper.

---

> > > > ### Author Response · Authors · 2025-08-03
> > > > **Incorporating Additional Results, Related Works, and Limitations in Final Revision**
> > > >
> > > > Dear Reviewer erLh,
> > > >
> > > > Thank you very much for your kind response and for raising the score. We truly appreciate it.
> > > >
> > > > We will make sure to incorporate additional experimental results, discussions of related works and limitations into the final version of the paper.
> > > >
> > > > Thank you again for your constructive feedback and support!
> > > >
> > > > Best regards,
> > > > Authors of Paper #2041

---

### Official Review · Reviewer_Y7QF · 2025-06-30

**Clarity:** 2
**Significance:** 3
**Originality:** 3
**Rating:** 4
**Confidence:** 1

**Summary:**

This paper proposes a method named SceneDesigner, which achieves precise control of 9D poses (position, size, orientation) in multi-object images. The author introduced the CNOCS mapping as the core representation, enhancing the interpretability and training efficiency of spatial control. To support the training, the authors constructed the ObjectPose9D dataset and designed a two-stage training mechanism to alleviate the problem of uneven pose distribution. The experimental results show that this method significantly outperforms the existing methods in both single-object and multi-object generation tasks. The overall work is innovative and practical.

**Questions:**

- This method currently cannot precisely control the specific shape of the object and only supports approximate representations based on cubes. May I ask if the author has considered combining explicit shape modeling (such as point cloud or mesh) to enhance the expression ability of detailed geometry?

- In multi-object scenarios, as the number of semantic concepts in text prompts increases, the model is prone to problems such as insufficient generation and attribute confusion. Although the decoupled sampling strategy is introduced for mitigation, its computational cost is relatively high. Has the author considered a more lightweight reasoning mechanism to control the reasoning overhead while ensuring consistent alignment of multiple objects?

**Ethical Concerns:**

["NO or VERY MINOR ethics concerns only"]

**Final Justification:**

I'm not familiar with this topic, so I'll keep the score.

**Limitations:**

yes

**Quality:**

3

**Strengths And Weaknesses:**

Strengths:

- The method is innovative and for the first time realizes the joint control of the 9D posture (position, size, orientation) of multiple objects.

- The CNOCS representation was proposed, which takes into account both universality and geometric interpretability and is applicable to multiple categories.

- A high-quality ObjectPose9D dataset was constructed, making up for the deficiency of existing public data in multi-object pose annotation.

- The experimental design is comprehensive. Both quantitative indicators and user surveys show that the method is significantly superior to the existing methods in terms of controllability and image quality.

Weaknesses:

- The analysis of the limitations of the existing methods is somewhat one-sided, and the selection of comparative methods is also relatively limited.

- How is the result evaluation of real-world/extraterritorial objects？

- How is the computational cost considered?

---

> ### Author Rebuttal · Authors · 2025-07-31
>
> ## W1(a). Analysis of the existing methods
> Thanks for your comments! We have included a detailed analysis of the limitations for the related methods in Appendix A.2. In this section, we compare existing approaches from multiple perspectives, including: (1) Controllability of 9D poses: We discuss the limitations of each method in terms of controllability, such as LOOSECONTROL [6] being unaware of orientation, Continuous 3D Words [8] being restricted to a 180° frontal angle range and rigid layouts, etc. (2) Dataset and training strategy: We analyze how the training data (e.g., synthetic vs. real-world) and learning strategies in prior works affect their performance. We will add more analysis of the limitations in the revised version.
>
> ---
>
> ## W1(b). Limited comparative method
> We acknowledge that the selected methods might seem limited. This is primarily because the field of 9-DoF multi-object pose control is still an emerging area and there are only a few methods in this field.
>
> As detailed in the Section 4.1 of our paper, the most recent and relevant works, such as ORIGEN[30] and Compass Control[36], were not open-sourced during our research. This prevented us from conducting a broader comparison. Consequently, we chose LOOSECONTROL (LC) and Continuous 3D Words (C3DW) for comparison since they are representative and open-source works in the field of 3D-aware control.
>
> ---
>
> ## W2 Evaluation of real-world/extraterritorial objects
> Thanks for your valuable comment! Our model exhibits strong generalization ability in real-world/extraterritorial objects beyond its training distribution. For example, the method successfully generates high-fidelity images for various objects, such as the capybara, tiger and rhinoceros in Figure 8, as well as some movie characters in Figure 10 and Figure 11 from our paper. Our training data is constructed from datasets like OmniNOCS [19] and MS-COCO [24], whose annotations do not include these categories. Furthermore, during our process of inspecting and filtering the dataset, we did not find the annotated instances of these objects. This provides strong evidence that our model effectively generalizes to a broad range of unseen concepts.
>
> ---
>
> ## W3 Computational cost
>
> We have provided a detailed introduction of required computational resources in Section 4. Concretely, the training process was conducted on 6 NVIDIA A800 80G GPUs, and the model was optimized with 45K and 5K steps in the first and the second stage, respectively.
>
> We have also evaluated the computational cost for generating images with varying numbers of objects on a single A800 GPU. The results are presented in the Table provided in our response to "Question 2".
>
> ---
>
> ## Q1 Combining explicit shape to enhance the expression of the detailed geometry
> Thanks for your insightful comments. We agree that a precise shape representation can improve the geometric accuracy of generated objects. However this could result in a less user-friendly experience for 9-DoF pose control, as users need to find 3D shapes for different object categories. Our primary motivation was to avoid users providing such cumbersome inputs. Therefore, we extend the NOCS to CNOCS variant, using a 3D bounding box abstraction instead of the precise object surface. The proposed CNOCS map achieves both user-friendly interaction and precise pose control.
>
> ---
>
> ## Q2 Lightweight technology for multi-object generation
> Thank you for your valuable suggestion regarding the computational cost of multi-object generation. As mentioned in the "Limitations" section, although the proposed DOS (Disentangled Object Sampling) technique mitigates the insufficient generation and attribute leakage issues, it introduces additional computation. We have conducted an analysis to quantify the overhead. The following Table details the inference time and VRAM usage on a single NVIDIA A800 GPU for generating images with 1, 2 and 3 objects.
>
> | Number of Objects | Inference Time (s) | VRAM Usage (GB) |
> |-------------------|--------------------|-----------------|
> | 1                 | 3.3                | 24.6            |
> | 2                 | 5.1                | 26.1            |
> | 3                 | 6.4                | 29.2            |
>
> As shown in the Table above, both inference time and VRAM usage increase as the number of objects grows. In our future work, we plan to explore more lightweight mechanisms to further reduce the computational cost, such as incorporating additional supervision during training to resolve object entanglement.
>
> ---

---

> > ### Comment · Reviewer_Y7QF · 2025-08-05
> >
> > Dear Authors
> >
> > Thank you for your efforts, author. However, I'm not familiar withthis topic. I have tried my best to review your article and other reviewers again, but I can only offer some suggestions that interest me. So I will keep the score.
> >
> > Best regards,
> >
> > Reviewer Y7QF

---

> > > ### Author Response · Authors · 2025-08-05
> > > **Thank you for your time and comments**
> > >
> > > Dear Reviewer Y7QF,
> > >
> > > Thank you very much for taking the time to review our paper and for participating in the discussion. We truly appreciate your suggestions and thoughtful consideration.
> > >
> > > If there are any remaining questions or points you’d like to further discuss, we would be happy to clarify.
> > >
> > > Best regards,
> > > Authors of Paper #2041

---

### Official Review · Reviewer_5SeK · 2025-07-01

**Clarity:** 2
**Significance:** 2
**Originality:** 2
**Rating:** 4
**Confidence:** 1

**Summary:**

The authors tackle the problem of conditional image generation, and focus on adding controllability by explicitly modelling the object 9 DoF pose information as an additional input to the model. In order to train the model, a new dataset is proposed, consisting of the Microsoft COCO dataset with additional annotations of 9 DoF  bounding boxes of the objects. Some tricks during training are proposed to improve the generation results. Comparisons are provided against existing approaches with a user study and some ablation experiments.

**Questions:**

* It seems like most of the failure cases of LOOSECONTROL are from the model trying to generate box shapes. Is this due to the default parameters?
* Are approaches compared to trained with the same data?
* How does a baseline with pose information as text compare against the existing approach, e.g., generating something for "An elephant on the grassland facing left" vs providing the pose as is currently done.

**Ethical Concerns:**

["NO or VERY MINOR ethics concerns only"]

**Final Justification:**

The authors have answered my questions and I still lean towards  accept. The paper could still benefit from reorganizing and rewriting parts of the paper as indicated in my original review.

**Limitations:**

The limitations discussion in the appendix should probably be moved to the main paper.

**Paper Formatting Concerns:**

None.

**Quality:**

2

**Strengths And Weaknesses:**

Strengths:
* Additional dataset annotations for the 9 DoF generation task.
* Compelling results when compared to existing approaches such as LOOSECONTROL.
* Elegantly handles the multi-object case.
* User study comparing with existing approaches.

Weaknesses:
* User study doesn't decouple different aspecs such as quality, pose fidelity, and text-to-image-alignment.
* Some important details are put in the appendix instead of the main paper.
* No significant technical contribution.

Other comments:
* User study should probably be in the main paper and not an appendix as it is mentioned in the experiments text.
* Similarly, limitations and discussion should probably be included in the main paper instead of the appendix.

---

> ### Author Rebuttal · Authors · 2025-07-31
>
> ## W1 User study doesn't decouple different aspects
> Thanks for your comment regarding our user study. We would like to clarify that our user study was designed to evaluate different aspects of the compared methods. As shown in Table 4 from our paper, we asked participants to assess the generated images from the following perspectives: Image Quality, Location Fidelity, Size Fidelity, Orientation Fidelity, and Text-to-image Alignment. This provides a comprehensive and fine-grained evaluation of different methods.
>
> ---
>
> ## W2 Suggestions for reorganizing
> We appreciate your suggestion regarding the organization of our paper. In the revised version, we will move important details (such as user study, limitations, and discussion) from the appendix to the main paper.
>
> ---
>
> ## W3 No significant technical contribution
> Thanks for your comment! Our main technical contribution is extending the ordinary NOCS map to the CNOCS variant. Existing NOCS-annotated datasets are limited to a few categories, and estimating NOCS maps for in-the-wild images is highly susceptible to noisy geometric estimation (e.g., point clouds and object masks). These challenges prevent NOCS maps from being successfully used for pose-controlled generation.
>
> In contrast, our CNOCS map utilizes a 3D bounding box abstraction instead of the precise object surface, making it more robust to inaccurate geometric estimations. This advantage enables our work to be the first to successfully leverage a NOCS-like representation for 9-DoF pose control task. Furthermore, since the CNOCS map does not require precise object shapes, it offers better user-friendliness than the NOCS map.
>
> ---
>
> ## Q1 The generated box-shaped objects in LOOSECONTROL
> We conducted extensive tests on LOOSECONTROL (LC) [6] using both its default parameters and other meaningful parameter settings, and we found that it generates box-shaped objects most of the time. We list some possible reasons below:
> * **Limitations of the Training Strategy**: LC employs LoRA to fine-tune a pre-trained depth-conditioned ControlNet. The number of trainable parameters (0.2024% of the total) and training steps (200) are insufficient to fully change the base model's inherent bias, where the structure of the generated image strictly conforms to the input depth map (e.g., a box-shaped depth map results in a box-shaped object).
> * **The Bias of the Training Data**: LC's training data is primarily composed of indoor scenes, leading to poor performance on general scenarios.
>
> ---
>
> ## Q2 Are approaches compared to trained with the same data
> For compared methods like LOOSECONTROL [6] and C3DW [8], we used the official checkpoints provided by the authors without any additional training.
>
> In our ablation studies, we conducted controlled experiments to assess the impacts of the proposed components. The models corresponding to the I-CNOCS and C-CNOCS variants were trained with our full dataset. The model for the "w/o MS-COCO" setting was only trained with the OmniNOCS [19] part of our dataset to demonstrate the importance of the constructed training data from MS-COCO [24].
>
> ---
>
> ## Q3 The baseline with pose information as text
> Thank you for this valuable question. We conducted an ablation study where the model is solely conditioned on the pose description. In this setting, we convert the 9D pose into descriptive text using predefined rules and templates (e.g., "on the left of the image, the size is large, viewed from the back") and append it to the original prompt. We evaluated this method on the ObjectPose-Single benchmarks and the quantitative results of this experiment are presented in the Table below.
>
> | Setting           | $Acc_{ls}$ (%)↑ | mIoU (%)↑ | Abs.Err ↓ | $Acc$\@22.5° (%)↑ |
> | ----------------- | --------------------- | --------- | --------- | -------------- |
> | Prompt only       | 12.90                 | 14.32     | 88.43     | 25.31          |
> | **SceneDesigner (ours)** | 51.12                 | 58.55     | 14.87     | 87.10          |
>
> We found that controlling the object pose solely with textual descriptions is insufficient for achieving accurate and reliable results. While this approach can successfully generate images for some common poses (e.g., "an elephant facing left" or "an elephant facing front"), it fails to understand the descriptions that contain precise or complex poses. For example, when providing prompts to describe precise orientations (e.g., "40 degrees" or "65 degrees") or some back-facing views ("an elephant facing backward"), the model struggles to generate objects with accurate poses.

---

> > ### Author Response · Authors · 2025-08-05
> > **Thank you and open to further discussion**
> >
> > Dear Reviewer 5SeK,
> >
> > Thank you again for your time and effort in reviewing our paper. We sincerely appreciate your comments and suggestions.
> >
> > We would be happy to clarify or elaborate on any points that may help further discussion or assist in your evaluation. Please feel free to reach out during the discussion period if you have any questions or thoughts.
> >
> > Best regards,
> > Authors of Paper #2041

---

> > > ### Comment · Reviewer_5SeK · 2025-08-06
> > >
> > > My concerns have been mainly addressed. I thank the authors for the rebuttal.

---

> ### Author Response · Authors · 2025-08-06
> **Thank you for your time and comments**
>
> Dear Reviewer 5SeK,
>
> Thank you very much for taking the time to review our paper and for participating in the discussion. We truly appreciate your suggestions and thoughtful consideration.
>
> If there are any remaining questions or points you’d like to further discuss, we would be happy to clarify.
>
> Best regards,
> Authors of Paper #2041

---

### Official Review · Reviewer_ia38 · 2025-07-01

**Clarity:** 3
**Significance:** 2
**Originality:** 2
**Rating:** 5
**Confidence:** 4

**Summary:**

This paper aims to address the task of controllable image generation using 9D poses of single/multiple objects as conditions. The authors propose the CNOCS map, which projects 3D bounding box information of objects into image space to encode size, location, and orientation for improved controllability. Moreover, they construct a dataset with 9D pose annotations. They also introduce several strategies to improve the controllability and fidelity of their method, such as fine-tuning with rewards and Disentangled Object Sampling during inference time. The qualitative and quantitative results show the effectiveness of their approach.

**Questions:**

Please refer to the weaknesses.

**Ethical Concerns:**

["NO or VERY MINOR ethics concerns only"]

**Final Justification:**

The authors’ detailed rebuttal has clearly addressed my concerns and questions. They propose a valid method for controllable image generation based on 9D pose, and their experiments validate the proposed representation. However, the paper would also benefit from improving its organization before release. Considering its novelty, completeness, and performance, I would raise my score to Accept.

**Limitations:**

yes

**Paper Formatting Concerns:**

no major formating issues

**Quality:**

2

**Strengths And Weaknesses:**

Strengths:

S1: This paper addresses an interesting problem of 9D pose-guided image generation, which is yet to be fully explored in the community.

S2: The results of this work are strong and impressive compared to the baselines.

S3: The authors define a comprehensive and complete set of evaluation metrics for this specific task.

S4: This paper is clearly written and easy to follow.

Weaknesses:

W1: The idea of the CNOCS map representation is simple, as it is a straightforward extension of the NOCS map from [48]. The CNOCS map only considers the 3D bounding box instead of the object surface. I suspect this coarse representation may lead to the generated object having a box-like shape, similarly as seen in the multi-object results of LC in Figure 5. It would be better if the authors conducted experiments comparing the CNOCS and NOCS representations on this task. Additionally, how does the model perform without CNOCS maps? Experimental results of only replacing CNOCS maps with other representations, such as depth maps or 2D rendering of 3D bounding boxes, in the proposed pipeline would better support the effectiveness of the CNOCS map.

W2: I am a bit confused about the role of the three variants of CNOCS maps based on different encoding functions, as described in Section 3.3. They are all derived from the proposed CNOCS map, with the only difference being in the final steps. Under this assumption, their performance should be similar. However, from Table 3 and Figure 6, I notice a significant performance difference between C-CNOCS and I-CNOCS maps. I am curious about the reason for this difference, considering they are all variants of your proposed CNOCS map.

W3: I have some suggestions regarding the organization of Section 3.3. The part explaining Equation 2 is redundant, as the text description sufficiently explains the proposed representation. Also, it would be better to introduce the other two variants of the CNOCS map in the experiment section for comparison, rather than in the method section.

---

> ### Author Rebuttal · Authors · 2025-07-31
>
> ## W1.(a) The CNOCS map is a straightforward extension of the NOCS map
> Thanks for your feedback! We agree that the CNOCS map is a variant of the NOCS map.
>
> Existing datasets with NOCS map annotations are limited to a few categories, while estimating the NOCS maps for in-the-wild images is susceptible to the noisy geometric estimation (e.g., point clouds and object masks). Consequently, NOCS maps have not yet been successfully applied to pose-controlled generation. In contrast, our CNOCS map uses a 3D bounding box abstraction instead of the precise object surface, making it more robust to inaccurate geometric estimation. This advantage enables our work to be the first to use a NOCS-like representation for high-fidelity, multi-object 9-DoF pose control. Furthermore, since the CNOCS map does not require precise object shapes, it offers better user-friendliness than the NOCS map.
>
> ---
>
> ## W1.(b) Concern about generating box-shaped objects
> Thanks for your valuable comment! To be frank, no box-shaped objects were observed in the images generated from our method. The motivation for using a 3D bounding box abstraction instead of the precise object surface is to avoid cumbersome 3D shapes provided by users. For the failure cases of LOOSECONTROL (LC) [6] shown in Figure 5 from our main paper, we list some possible reasons below.
> * **Limitations of the Training Strategy**: LC employs LoRA to fine-tune a pre-trained depth-conditioned ControlNet. The number of trainable parameters (0.2024% of the total) and training steps (200) are insufficient to fully change the base model's inherent bias, where the structure of the generated image strictly conforms to the input depth map (e.g., a box-shaped depth map results in a box-shaped object).
> * **The Bias of the Training Data**: LC's training data is primarily composed of indoor scenes, leading to poor performance in general scenarios.
>
> In contrast to LC, our method trains a CNOCS-conditioned ControlNet from scratch, with more learnable parameters and without the inherent bias of pre-trained models. Additionally, our ObjectPose9D dataset covers a broader range of objects and scenes compared to the dataset used in LC. Therefore, our method avoids the generation of box-shaped objects.
>
> In addition, we have conducted experiments comparing the CNOCS and NOCS representations in the following response.
>
> ---
>
> ## W1.(c) Ablation studies against alternative representations
> Thanks for your valuable comments. We have conducted the following experiments to further illustrate the effectiveness of the CNOCS map. We evaluated these methods on the ObjectPose-Single benchmarks, and the comparison results are presented in the Table below. For the sake of fairness, we only conducted one-stage training (i.e., without reinforcement learning) for all the models, and compared them with the results of our first-stage model ("**ours, stage 1**" in the "Setting" column).
>
> * **Comparison with NOCS Map**: Based on the training data used in our method, we additionally constructed NOCS maps through estimated point clouds and orientations, and subsequently trained a model using this representation. As demonstrated in the Table below, the NOCS-based model (denoted as "NOCS map" in the "Setting" column) achieves better accuracy in location and size than our model. This improvement stems from the fact that the object mask is more precise in representing the shape. However, the orientation accuracy of the NOCS-based model is lower than that of our approach. This is because the NOCS map is more susceptible to the noisy geometric estimation (e.g., object masks, point clouds), resulting in inaccurate annotation, thereby degrading model performance. In contrast, our CNOCS map employs 3D bounding box abstraction rather than precise object surface, which not only eliminates the need for cumbersome 3D shape inputs but also enhances robustness to the noisy geometric estimation.
>
>
> * **Ablation without CNOCS Map**: We conducted an ablation study where the CNOCS map was removed, relying solely on text prompts for pose guidance. Concretely, we transfer the input pose to the description based on rules and prompt templates, like "on the left of the image, the size is large, viewed from the back", which is then appended to the original prompt. The Table below shows that the model (denoted as "Prompt only" in the "Setting" column) exhibits poor fidelity with the input condition, which highlights the necessity of the CNOCS map for fine-grained orientation control.
>
> * **Comparison with Depth Maps**: For fairness, we additionally trained a depth-conditioned ControlNet with our dataset from scratch. The Table below indicates that our CNOCS map outperforms the depth map (denoted as "Depth map" in the "Setting" column) in orientation accuracy. This is because the depth map does not explicitly encode orientation information and lacks the details of object appearances, resulting in poor fidelity.
>
>
> * **Comparison with 2D Rendering of 3D Bounding Boxes**: Similar to the above experiments, we additionally trained a ControlNet with our dataset from scratch, which is conditioned on the 2D rendering of 3D bounding boxes. The Table below (denoted as "2D rendering" in the "Setting" column) shows that the model also exhibits a poor performance in orientation accuracy, since the representation lacks orientation information.
>
> | Setting           | $Acc_{ls}$ (%)↑ | mIoU (%)↑ | Abs.Err ↓ | $Acc$\@22.5° (%)↑ |
> | ----------------- | --------------------- | --------- | --------- | -------------- |
> | NOCS map          | 50.12                 | 56.82     | 48.70     | 44.91          |
> | Prompt only       | 12.90                 | 14.32     | 88.43     | 25.31          |
> | Depth map         | 47.89                 | 55.59     | 75.21     | 38.96          |
> | 2D rendering      | 41.94                 | 47.09     | 81.66     | 32.26          |
> | **ours, stage 1** | 43.18                 | 50.32     | 43.85     | 52.36          |
> | **ours, stage 2** | 51.12                 | 58.55     | 14.87     | 87.10          |
>
> ---
>
> ## W2 The performance gap between the CNOCS variants
> Thank you for this insightful comment. Here, we explain the reasons for the performance gap between the C-CNOCS and I-CNOCS variants.
>
> The C-CNOCS map provides a coarse and spatially-consistent signal. Specifically, a constant vector (Euler angle of object orientation) is assigned to each pixel within the object area. This provides a unified guidance for all pixels, making it challenging to learn to generate various object appearances under different orientation conditions.
>
> In contrast, the I-CNOCS map provides dense and spatially varying signals. Concretely, each pixel within the object area is assigned its corresponding normalized 3D coordinates in the object space. Compared to the C-CNOCS map, this variant facilitates the network to capture the relationship between object appearance and the 3D structure, thereby better encoding the orientation information and making the training process more efficient and stable.
>
> ---
>
> ## W3 Suggestions for reorganizing the paper
> Thanks for your valuable suggestions on improving the organization and clarity of our paper.
>
> * **The Redundancy of Equation 2**.
> We presented Equation 2 to formalize the construction process of the CNOCS map, supplemented by textual explanations. We acknowledge your concern about potential redundancy and will carefully revise this part to make it more concise and clear.
>
> * **The Statement of CNOCS map Variants**.
> We agree with your suggestion and will move the discussion of the CNOCS map variants (e.g., I-CNOCS and C-CNOCS maps) to the experiment section.

---

> > ### Author Response · Authors · 2025-08-05
> > **Thank you and open to further discussion**
> >
> > Dear Reviewer ia38,
> >
> > Thank you again for your time and effort in reviewing our paper. We sincerely appreciate your comments and suggestions.
> >
> > We would be happy to clarify or elaborate on any points that may help further discussion or assist in your evaluation. Please feel free to reach out during the discussion period if you have any questions or thoughts.
> >
> > Best regards,
> > Authors of Paper #2041

---

> ### Comment · Reviewer_ia38 · 2025-08-05
>
> Dear Authors,
>
> Thank you for your detailed explanations. I have carefully read all your comments. Your responses provide strong support for the effectiveness of your proposed representation. However, I still have one more concern, which may potentially affect my score.
>
> Although the NOCS map is sensitive to noisy point clouds, it still contains rich information about the shape of objects, which could better encode their corresponding orientation information. Thus, if they all use the same training strategies and are trained on the same dataset, the NOCS map should theoretically perform better than the CNOCS map. However, in your experiments, it performs worse in Abs. Err and Acc@22.5%. It would be helpful if you could provide more in-depth analysis on this.
>
> Moreover, considering user-friendly interactivity, the NOCS map might also be a better choice than the CNOCS map if the control condition involves drawing sketches or strokes. Therefore, I think both methods are user-friendly, depending on different application scenarios.
>
> Best,
>
> ia38

---

> ### Author Response · Authors · 2025-08-06
> **More in-depth analysis**
>
> ## More in-depth analysis of the orientation accuracy
> Thank you for your insightful observations! After carefully reading your suggestions, we have conducted a comprehensive analysis of the images generated by both the NOCS-based model and our model.
>
> Visually, for the NOCS and CNOCS maps representing the same 9-DoF pose (i.e., predicted from the same image), the objects generated by two models have similar overall orientation but differ in local regions. For example, for the object generated from an NOCS map, certain body parts (such as a person's limbs or head) exhibit local deformations based on the object mask.
>
> In addition to the noisy geometric predictions, we attribute the differences of orientation accuracy shown in the Table from W1.(c) to the varying accuracy of the estimation model (Orient Anything) across different scenarios. When we manually filtered the dataset, we observed that Orient Anything exhibits lower accuracy for objects with complex local deformations compared to those with simple postures (e.g., most body parts are oriented in the same direction). Therefore, the orientations predicted by Orient Anything on partial images generated by the NOCS-based model exhibit some error, since most of these instances involve complex non-rigid deformations (e.g., an athlete swinging a baseball bat, where the head, body, and arms have different orientations). In contrast, our CNOCS-based model typically produces objects with consistent orientations across body parts, leading to better quantitative performance in orientation accuracy.
>
> As mentioned in the "Limitations" section of our paper, the CNOCS variant can not control the precise shape of the object, where the NOCS map is more effective. However, as explained in the response below, the CNOCS map offers a more user-friendly interaction for the 9-DoF pose control task, which is the topic of our paper. Nevertheless, we acknowledge that improving the model's capabilities in shape control is valuable, and we plan to address the challenges of this application in our future work.
>
> ---
>
> ## The interactivity of the NOCS map and CNOCS variant
> For constructing the NOCS map, using the 2D sketches instead of the object's 3D shape may lead to the following potential risks.
>
> * **Lack of 3D Information to Construct the NOCS Map**: The sketches and strokes do not contain 3D information like the depth and orientation, making it difficult to infer the normalized 3D coordinates of pixels to construct the NOCS map. However, introducing 3D information (e.g., depth map) would burden the users.
>
> * **Inaccurate 2D Object Masks**: While sketches and strokes are user-friendly and effective for shape control in some simple orientations (e.g., front-, back-, left-, or right-facing objects), the complex 9-DoF pose requires the user to imagine the object's appearance from the camera view. For example, inferring the object's silhouette from an elevated and oblique viewpoint is challenging. As a result, it is difficult to obtain the expected results with the imprecise object mask provided by users. In contrast, the CNOCS map only requires the user to place 3D bounding boxes in the 3D space (as shown in Figure 2 of our main paper), which alleviates the challenges.
>
>
>
> Nevertheless, we believe this is a very important and valuable research direction. If users can control the object's shape by simply drawing the sketches, it would lead to a better user experience in 9-DoF pose control.

---

### Note · Authors · 2025-08-13

We sincerely thank the reviewers for their valuable time and insightful suggestions. We are encouraged that the reviewers recognized the novelty and significance of our work, as well as the following key contributions.
* The paper addresses an interesting problem of 9D pose control in image generation, which has not been fully explored (ia38, Y7QF).
* The proposed method (SceneDesigner) significantly outperforms prior works in both controllability and image quality (ia38, 5SeK, Y7QF, erLh).
* The proposed CNOCS map is an efficient and novel representation for 9D pose control (Y7QF, erLh).
* The high-quality ObjectPose9D dataset and the proposed pipeline for annotating 9D poses are valuable (5SeK, Y7QF, erLh).
* The proposed techniques are reasonable and effective to improve the controllability and fidelity, such as the reward optimization and DOS algorithm (ia38, 5SeK, erLh).
* The experimental design and evaluation metrics are comprehensive (ia38, 5SeK, Y7QF).
* This paper is clearly written and easy to follow (ia38).

We address the main concerns raised by the reviewers:
* To further demonstrate the effectiveness of the CNOCS map (ia38, 5SeK, erLh), we have conducted additional experiments and compared our CNOCS map with alternative representations, including pose embedding, text-only guidance, depth map, and standard NOCS map. The results validate the effectiveness of our CNOCS map.
* We have conducted additional analyses to demonstrate that CNOCS maps are more user-friendly for 9D pose control than explicit shape modeling (e.g., point clouds or NOCS maps), while achieving comparable pose accuracy (ia38, Y7QF). While the NOCS map offers better shape control, it requires users to provide precise 3D models. In contrast, users only need to manipulate 3D boxes with our CNOCS variant.
* We have conducted extensive experiments and additional analyses to confirm that our method does not generate box-like objects as observed in LOOSECONTROL (ia38, 5SeK).

We are committed to making the following improvements:
* Incorporating the new experimental results, discussions, and limitations into the revision.
* Refining the organization as suggested, carefully checking and correcting typos.
* Publicly releasing the ObjectPose9D dataset, along with the source code and trained weights.

Finally, we thank the reviewers and AC for their time and suggestions. We believe our work is valuable for 9D pose control task.

---

### Decision · Program_Chairs · 2025-09-17

**Decision:**

Accept (spotlight)

**Comment:**

The paper introduces the model for manipulating object poses in 9 DOF in image generative models. The authors use bounding boxes to control the object pose.

The reviewers highlight the strong contributions of the paper, specifically that 3D controllability is essential missing piece for image generation. Empirically, the paper presents strong examples on complex scene control and compositionality. Throughout the rebuttal, the authors clarified the remaining concerns over clarity, limited evaluations and ablations.

In the final version, the authors are encouraged to clarify the  difference to a related work Neural Assets that provides a similar capability, as well as addressing the potential misuse scenarios and providing a stronger mitigation plan for those.